# Mitochondrial Bol1 and Bol3 function as assembly factors for specific iron-sulfur proteins

Marta A Uzarska[1†], Veronica Nasta[2†], Benjamin D Weiler[1†], Farah Spantgar[1], Simone Ciofi-Baffoni[2,3], Maria Rosaria Saviello[2,3], Leonardo Gonnelli[2,3], Ulrich Mühlenhoff[1], Lucia Banci[2,3*], Roland Lill[1,4*]

[1]Institut für Zytobiologie und Zytopathologie, Philipps-Universität, Marburg, Germany; [2]Magnetic Resonance Center CERM, University of Florence, Florence, Italy; [3]Department of Chemistry, University of Florence, Florence, Italy; [4]LOEWE Zentrum für Synthetische Mikrobiologie SynMikro, Marburg, Germany

**Abstract** Assembly of mitochondrial iron-sulfur (Fe/S) proteins is a key process of cells, and defects cause many rare diseases. In the first phase of this pathway, ten Fe/S cluster (ISC) assembly components synthesize and insert [2Fe-2S] clusters. The second phase is dedicated to the assembly of [4Fe-4S] proteins, yet this part is poorly understood. Here, we characterize the BOLA family proteins Bol1 and Bol3 as specific mitochondrial ISC assembly factors that facilitate [4Fe-4S] cluster insertion into a subset of mitochondrial proteins such as lipoate synthase and succinate dehydrogenase. Bol1-Bol3 perform largely overlapping functions, yet cannot replace the ISC protein Nfu1 that also participates in this phase of Fe/S protein biogenesis. Bol1 and Bol3 form dimeric complexes with both monothiol glutaredoxin Grx5 and Nfu1. Complex formation differentially influences the stability of the Grx5-Bol-shared Fe/S clusters. Our findings provide the biochemical basis for explaining the pathological phenotypes of patients with mutations in *BOLA3*.

*For correspondence: banci@ cerm.unifi.it (LB); lill@staff.uni-marburg.de (RL)

[†]These authors contributed equally to this work

Competing interests: The authors declare that no competing interests exist.

## Introduction

Mitochondria are essential organelles and are involved in numerous biological tasks including fatty acid degradation, citric acid cycle, respiration, ATP production, and the biogenesis of various protein cofactors such as heme, lipoic acid and iron-sulfur (Fe/S) clusters. Genetic defects in these pathways are associated with a spectrum of diverse mitochondriopathies with neurological, neurodegenerative, hematological, and metabolic phenotypes. Knowledge of the molecular deficits of the affected mitochondrial components is key for the development of a comprehensive understanding of mitochondrial diseases. A mutation in human *BOLA3* encoding a mitochondrial protein is associated with diverse defects summarized as multiple mitochondrial dysfunction syndrome 2 (MMDS2; *Baker et al., 2014*; *Cameron et al., 2011*; *Haack et al., 2013*). In particular, BOLA3 deficiency results in decreased functions of respiratory complexes I and II as well as lipoic acid-dependent enzymes such as pyruvate dehydrogenase (PDH), 2-ketoglutarate dehydrogenase (KGDH), and glycine cleavage system (GCS) (*Mayr et al., 2014*). BOLA3 was suggested to play a role in mitochondrial Fe/S protein biogenesis because of similar clinical and biochemical phenotypes in patients with mutations in the known ISC factors *NFU1* (causing MMDS1; *Cameron et al., 2011*; *Invernizzi et al., 2014*; *Navarro-Sastre et al., 2011*) and *IBA57* (causing MMDS3; *Ajit Bolar et al., 2013*; *Lossos et al., 2015*; for review see *Beilschmidt and Puccio, 2014*; *Stehling et al., 2014*). Another link for BOLA3 to mitochondrial Fe/S protein biogenesis is provided by the fact that related bacterial and plant Bol proteins interact with monothiol glutaredoxins, factors that play a critical role in Fe/S

**eLife digest** Proteins perform almost all the tasks necessary for cells to survive. However, some proteins, especially enzymes involved in metabolism and energy production, need to contain extra molecules called co-factors to work properly. In human, yeast and other eukaryotic cells, co-factors called iron-sulfur clusters are made in compartments called mitochondria before being packaged into target proteins.

Defects that affect the assembly of proteins with iron-sulfur clusters are associated with severe diseases that affect metabolism, the nervous system and the blood. Mitochondria contain at least 17 proteins involved in making iron-sulfur proteins, but there may be others that have not yet been identified. For example, a study on patients with a rare human genetic disease suggested that a protein called BOLA3 might also play a role in this process.

BOLA3 is closely related to the BOLA1 proteins. Here, Uzarska, Nasta, Weiler et al. used yeast to test how these proteins contribute to the assembly of iron-sulfur proteins. Biochemical techniques showed that the yeast equivalents of BOLA1 and BOLA3 (known as Bol1 and Bol3) play specific roles in the assembly pathway. When both of these proteins were missing from yeast, some iron-sulfur proteins – including an important enzyme called lipoic acid synthase – did not assemble properly. The experiments suggest that yeast Bol1 and Bol3 play overlapping and critical roles during the last step of iron-sulfur protein assembly when the iron-sulfur cluster is inserted into the target protein.

Lastly, Uzarska, Nasta, Weiler et al. used biophysical techniques to show how Bol1 and Bol3 interact with another mitochondrial protein that performs a more general role in iron-sulfur protein assembly. Defects in assembling iron-sulfur proteins are generally more harmful to human cells than yeast cells. Therefore, the next step is to investigate what exact roles BOLA1 and BOLA3 play in human cells and how similar this pathway is in different eukaryotes.

protein biogenesis [*Boutigny et al., 2013*; *Roret et al., 2014*; *Yeung et al., 2011*]). However, dedicated biochemical investigations of the molecular role of BOLA3 in mitochondria have not been reported hitherto.

Human BOLA3 belongs to a large protein family of bacterial origin (*Aldea et al., 1988*; *Willems et al., 2013*). Eukaryotes such as yeast and humans possess three BOLA family members that can be discriminated by conserved sequence elements (*Figure 1—figure supplement 1*). To date, the precise function of the BOLA proteins and the functional relationship between the three eukaryotic BOLA family members is unclear. Yeast Bol2 (formerly termed Fra2; [*Kumanovics et al., 2008*]) is cytosolic, and is involved in cellular iron regulation by forming heterodimeric, [2Fe-2S] cluster-containing complexes with the cytosolic monothiol glutaredoxins Grx3-Grx4 (*Li et al., 2009*; *Mühlenhoff et al., 2010*). The human relative BOLA2 also forms hetero-complexes with human GRX3 (*Banci et al., 2015*; *Li et al., 2012*), yet the exact physiological role of the Bol2 proteins is still poorly defined. Additionally to BOLA3, human mitochondria contain the homologous BOLA1 (*Willems et al., 2013*). Ablation of *BOLA1* in cultured human cells increases mitochondrial protein thiol oxidation and elicits alterations in mitochondrial morphology. BOLA1 (but not BOLA3) co-purified with tagged mitochondrial monothiol glutaredoxin GLRX5 that is crucial for Fe/S protein biogenesis (*Willems et al., 2013*; *Ye et al., 2010*). Both BOLA1 and BOLA3 have structural counterparts in many eukaryotes including yeast Bol1 and Bol3, respectively (*Figure 1—figure supplement 1*). Apart from the *BOLA3* patient cell analysis, no solid evidence for a direct role of the BOLA family proteins in cellular Fe/S protein metabolism has been described. We therefore sought to better define the physiological role of the mitochondrial Bol1-Bol3 (mBols) and cytosolic Bol2 proteins in this essential biosynthetic pathway by using the yeast *S. cerevisiae* as a model organism (*Beilschmidt and Puccio, 2014*; *Lill, 2009*; *Lill et al., 2012*; *Netz et al., 2014*; *Rouault, 2012*).

Mitochondrial Fe/S protein biogenesis in yeast involves 17 known ISC components which were inherited from bacteria and are conserved in eukaryotes (*Johnson et al., 2005*; *Lill, 2009*). The pathway can be divided into two major phases. First, components of the 'core ISC machinery' including the cysteine desulfurase Nfs1 and the scaffold protein Isu1 synthesize a [2Fe-2S] cluster, and transfer it transiently to the monothiol glutaredoxin Grx5 for subsequent assembly of mitochondrial [2Fe-2S]

proteins. Second, dedicated ISC factors use the Grx5-bound Fe/S cluster to assemble a [4Fe-4S] cluster and to facilitate its insertion into mitochondrial apoproteins such as aconitase, respiratory complexes, and the radical SAM Fe/S protein lipoic acid synthase (LIAS). Even though the molecular mechanisms of this late phase of Fe/S protein biogenesis are poorly resolved, recent studies have shown that the Isa1, Isa2 and Iba57 proteins cooperate to generate a [4Fe-4S] cluster (*Brancaccio et al., 2014*; *Gelling et al., 2008*; *Mühlenhoff et al., 2011*; *Sheftel et al., 2012*). Its insertion into specific target apoproteins is subsequently facilitated by Nfu1 and the P-loop ATPase Ind1, both transiently binding [4Fe-4S] clusters (*Bych et al., 2008*; *Sheftel et al., 2009*; *Tong et al., 2003*). The latter ISC protein is specific for maturation of respiratory complex I, but nothing is known about the molecular mechanisms underlying [4Fe-4S] cluster insertion by eukaryotic Nfu1 and Ind1.

While most proteins of the core ISC machinery are essential for cell viability, impairment of the ISC factors involved in the second phase of Fe/S protein biogenesis only compromises mitochondrial function of yeast cells. This striking difference for phases I and II is explained by the central, additional role of the core ISC machinery in both cytosolic Fe/S protein biogenesis and cellular iron regulation. Functional defects of core ISC factors lead to impaired assembly of essential Fe/S proteins such as nuclear DNA polymerases and helicases (*Gari et al., 2012*; *Netz et al., 2012*; *Stehling et al., 2012*) and an induction of the yeast iron regulon involving the above mentioned Bol2 (*Lill et al., 2012*; *Outten and Albetel, 2013*; *Paul and Lill, 2015*). Here, we employed a combination of cell biological, biochemical and ultrastructural methods to characterize the potential role of the Bol proteins in cellular Fe/S protein biogenesis and to define their position in the complex pathway. We also investigate the involvement of the mBols in cellular iron regulation in comparison to Bol2.

## Results

### Deficiency of Bol1-Bol3 causes defects in a subset of mitochondrial [4Fe-4S] enzymes

We first analyzed the sub-cellular localization of *S. cerevisiae* Bol1 (YAL044W-A) and Bol3 (formerly termed Aim1; (*Hess et al., 2009*) by cell fractionation. Both proteins were exclusively present in the mitochondrial fraction and absent in the cytosol (*Figure 1—figure supplement 2A*). Upon sub-fractionation of mitochondria by hypotonic swelling or detergent lysis, the Bol proteins behaved similar to the matrix proteins Tim44 and Mge1 (*Figure 1—figure supplement 2B*). Thus, Bol1 and Bol3 are constituents of the mitochondrial matrix. To examine the physiological function of mitochondrial Bol1-Bol3 (mBols), we created single and double deletion cells, as well as combined deletions with *BOL2* (*Supplementary file 1A*; strain background BY4742). All deletion strains grew at wild-type rates on minimal media containing the non-fermentable carbon sources glycerol or acetate with the exception of *bol13Δ* and *bol123Δ* strains (*Figure 1—figure supplement 3*). Apparently, simultaneous deletion of *BOL1* and *BOL3* created a weak respiratory defect which is slightly weaker than that seen for *NFU1* deletion (*Schilke et al., 1999*).

We then analyzed key mitochondrial enzyme activities to detect potential defects in Fe/S protein biogenesis, and we compared these results to data for a *NFU1* deletion strain which shows mild, yet specific Fe/S enzyme defects (*Navarro-Sastre et al., 2011*; *Schilke et al., 1999*). This analysis included the mitochondrial Fe/S proteins aconitase, succinate dehydrogenase (SDH or complex II), and yeast LIAS (Lip5). The latter can be assayed indirectly either by the enzyme activities or the lipoylation extent of pyruvate dehydrogenase (PDH) and 2-ketoglutarate dehydrogenase (KGDH) (*Gelling et al., 2008*; *Schonauer et al., 2009*). Further, we measured the activity of cytochrome *c* oxidase (COX). Even though this enzyme does not contain Fe/S clusters, it is frequently affected by mitochondrial Fe/S protein biogenesis defects, for unknown reasons (see, e.g., [*Mühlenhoff et al., 2011*; *Sheftel et al., 2012*]). All enzyme activities were normalized to malate dehydrogenase (MDH) that showed no significant changes in all these strains relative to wild type (not shown).

Single deletions of *BOL1* or *BOL3* did not affect any of these enzyme activities relative to wild-type cells with the exception of a slight, but significant decrease of SDH in *bol3Δ* cells (*Figure 1A–E*, bars 1–3). In contrast, double *BOL1-BOL3* deletions resulted in substantial, up to threefold decreases of the Fe/S cluster-dependent enzyme activities, an effect comparable to that seen for *nfu1Δ* cells (bars 4 and 6) (*Navarro-Sastre et al., 2011*). A notable exception was aconitase which retained

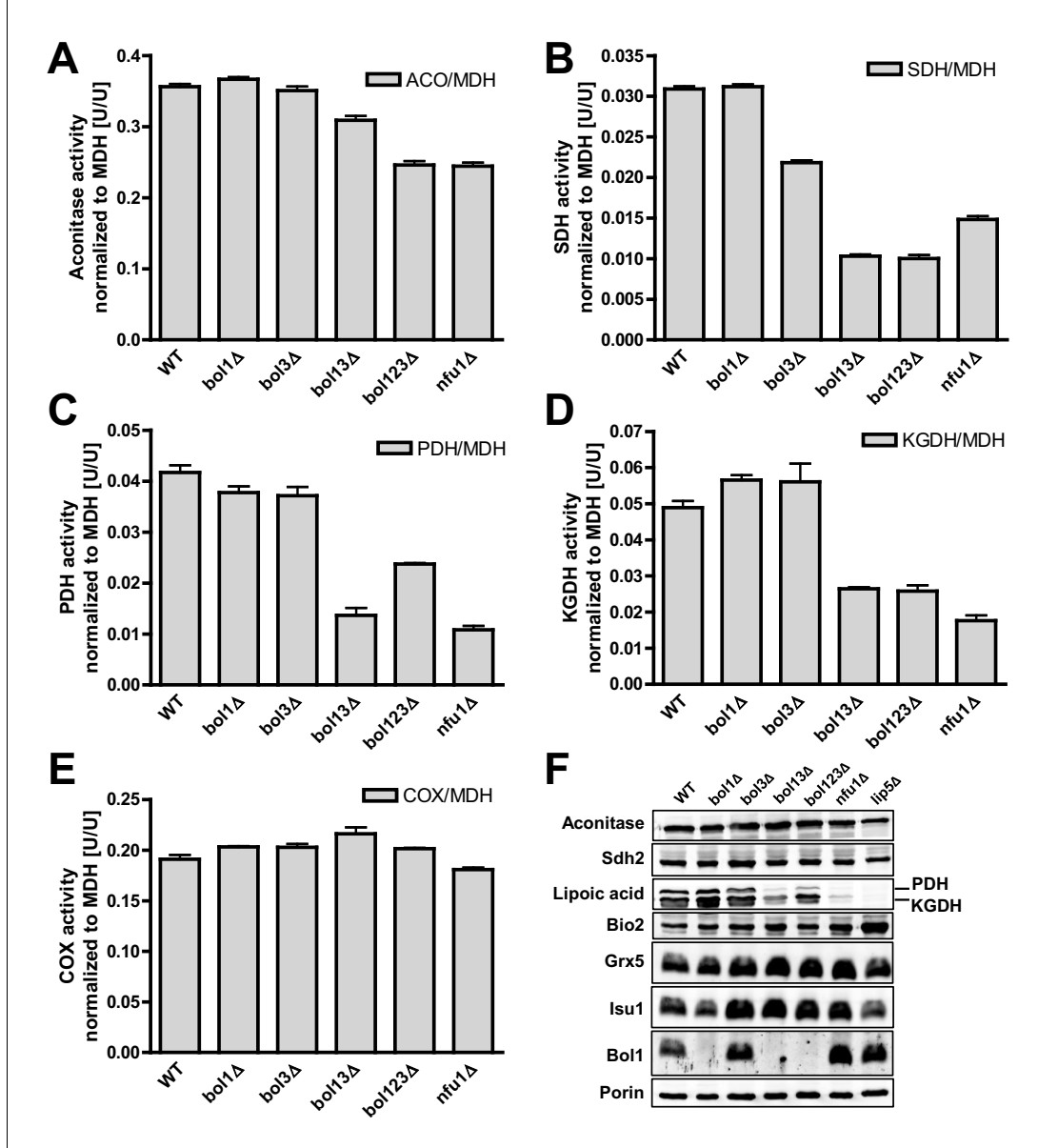

**Figure 1.** Deficiency of both mitochondrial Bol proteins causes defects in a subset of mitochondrial [4Fe-4S] enzymes. Wild-type (WT; strain BY4742), and the indicated *BOL* and *NFU1* deletion yeast strains were grown in minimal medium containing 2% galactose supplemented with 50 µM ferric ammonium citrate and used for the preparation of mitochondria. Mitochondrial extracts were assayed for specific activities of (**A**) aconitase (ACO), (**B**) succinate dehydrogenase (SDH), (**C**) pyruvate dehydrogenase (PDH), (**D**) 2-ketoglutarate dehydrogenase (KGDH), and (**E**) cytochrome *c* oxidase (COX). Values were normalized to those of malate dehydrogenase (MDH). Error bars indicate the SEM (n≥4). (**F**) Mitochondrial extracts were subjected to TCA precipitation and the levels of the indicated proteins and of lipoic acid attached to E2 subunits of PDH and KGDH were determined by immunostaining. Staining for porin served as a loading control.

The following figure supplements are available for figure 1:

**Figure supplement 1.** Cartoon of bacterial BolA and eukaryotic Bol1, Bol2 and Bol3 proteins.

**Figure supplement 2.** Bol1 and Bol3 are proteins of the mitochondrial matrix.

**Figure supplement 3.** The growth behavior of the various *BOL* gene deletion strains in comparison to *NFU1* deletion cells.

**Figure supplement 4.** Specific rescue of succinate dehydrogenase activities in *BOL* gene deletion cells by both mitochondrial Bol proteins.

*Figure 1 continued on next page*

*Figure 1 continued*

**Figure supplement 5.** Fe/S enzyme activity defects in cells lacking Bol1-Bol3 after growth in lactate medium.

normal activity in *bol13Δ* cells (*Figure 1A*). Thus, double but not single yeast *BOL* gene deletions created a similar phenotype as that observed in *BOLA3* patient cells (*Cameron et al., 2011*). Additional ablation of *BOL2* (strain *bol123Δ*, bar 5) did not substantially exacerbate these effects suggesting that the mitochondrial and cytosolic Bol proteins act independently. The decrease in SDH was likely due to a direct defect in Fe/S cluster assembly, because SDH protein levels remained unchanged in immunoblots (*Figure 1F*). In keeping with the decreased PDH and KGDH activities, we also found a lipoylation defect of the E2 subunits of these enzymes by immunostaining against lipoic acid (*Figure 1F*) (*Gelling et al., 2008*; *Schonauer et al., 2009*). All the observed effects were rather mild in comparison to defects reported for deletion mutants of other late-acting ISC factors such as Isa1, Isa2, and Iba57 (*Gelling et al., 2008*; *Mühlenhoff et al., 2011*). This notion was also evident from the lack of any effects on COX activity in all tested *BOL* deletion strains (*Figure 1E*). A severe COX diminution is observed in *ISA1, ISA2,* or *IBA57* mutants, yet the exact reason is unknown (*Gelling et al., 2008*; *Mühlenhoff et al., 2011*). In order to verify that the loss of the enzyme activities was specific, *BOL1* and *BOL3* genes were reintroduced into the *bol123Δ* deletion strain by expression from plasmids. Both Bol1 and Bol3 were able to efficiently rescue the deficiencies in SDH activities (*Figure 1—figure supplement 4*).

Previous studies on yeast ISC protein depletions, e.g., of the Isa proteins, had shown that the Fe/S protein defects were much more pronounced under strict respiratory conditions, e.g., upon growth in lactate medium (*Kispal et al., 1999*; *Mühlenhoff et al., 2011*) (see also *Figure 1—figure supplement 3*). Both aconitase and SDH activities of the various *BOL* deletion strains behaved similarly after growth in lactate- compared to galactose-containing media (*Figure 1—figure supplement 5*). In contrast, PDH and KGDH activities were much more severely (ten- vs. threefold) diminished in *bol13Δ, bol123Δ,* and *nfu1Δ* cells grown in lactate medium, similar to previous reports on Isa protein depletion (*Kaut et al., 2000*; *Mühlenhoff et al., 2011*). Together, these data show that the mBols were crucial for efficient maturation of LIAS, yet they play only an auxiliary role in SDH maturation. The levels of Bol1 and Nfu1 did not change or rather slightly decreased in the various deletion cells making a compensatory effect unlikely (data not shown; Bol3 levels could not be tested due to the lack of antibody detection). Obviously, Bol1 and Bol3 perform largely complementary roles, because only simultaneous deletion of both *BOL1-BOL3* genes created a significant LIAS defect.

## Bol1 and Bol3 are crucial for de novo Fe/S cluster maturation of specific mitochondrial [4Fe-4S] but not [2Fe-2S] proteins

To directly characterize the mBol involvement in mitochondrial Fe/S cluster assembly, we employed a well-established $^{55}$Fe radiolabeling and immunoprecipitation assay. This also allowed the testing of additional Fe/S proteins including such with [2Fe-2S] clusters. The various *BOL* deletion strains were radiolabeled with $^{55}$Fe, the Fe/S proteins were immunoprecipitated, and the precipitated radioactivity was quantified by scintillation counting (*Pierik et al., 2009*). Consistent with the enzyme activity data (*Figure 1*), we found no significant effect on $^{55}$Fe incorporation into aconitase upon single, double or triple *BOL* gene deletion (*Figure 2A*). For SDH (assayed by precipitation of the myc-tagged Fe/S cluster subunit Sdh2 co-expressed with Sdh1) a 40% decrease in $^{55}$Fe binding was observed upon deletion of *BOL3*, also in combination with *BOL1* and *BOL2* deletions (*Figure 2B*). Compared to other ISC gene deletions (see, e.g., *Gelling et al., 2008*; *Gerber et al., 2004*; *Mühlenhoff et al., 2011*; *Rodriguez-Manzaneque et al., 2002*; *Voisine et al., 2001*) this effect was rather weak. Similar, yet slightly stronger decreases in $^{55}$Fe binding (60%) were detected for mitochondria-targeted HiPIP, a bacterial [4Fe-4S] cluster-containing ferredoxin (*Figure 2C*). For all the tested Fe/S proteins, single deletion of *BOL1* had no effects on $^{55}$Fe binding. The observed defects were specific, because reintroduction of Bol1 or Bol3 into the *bol123Δ* strain via overproducing vectors completely restored wild-type $^{55}$Fe binding activity, whereas cytosolic Bol2 or Grx5, as a control, were ineffective (*Figure 2D*). Taken together, these data support a specific function of the mBols in the maturation

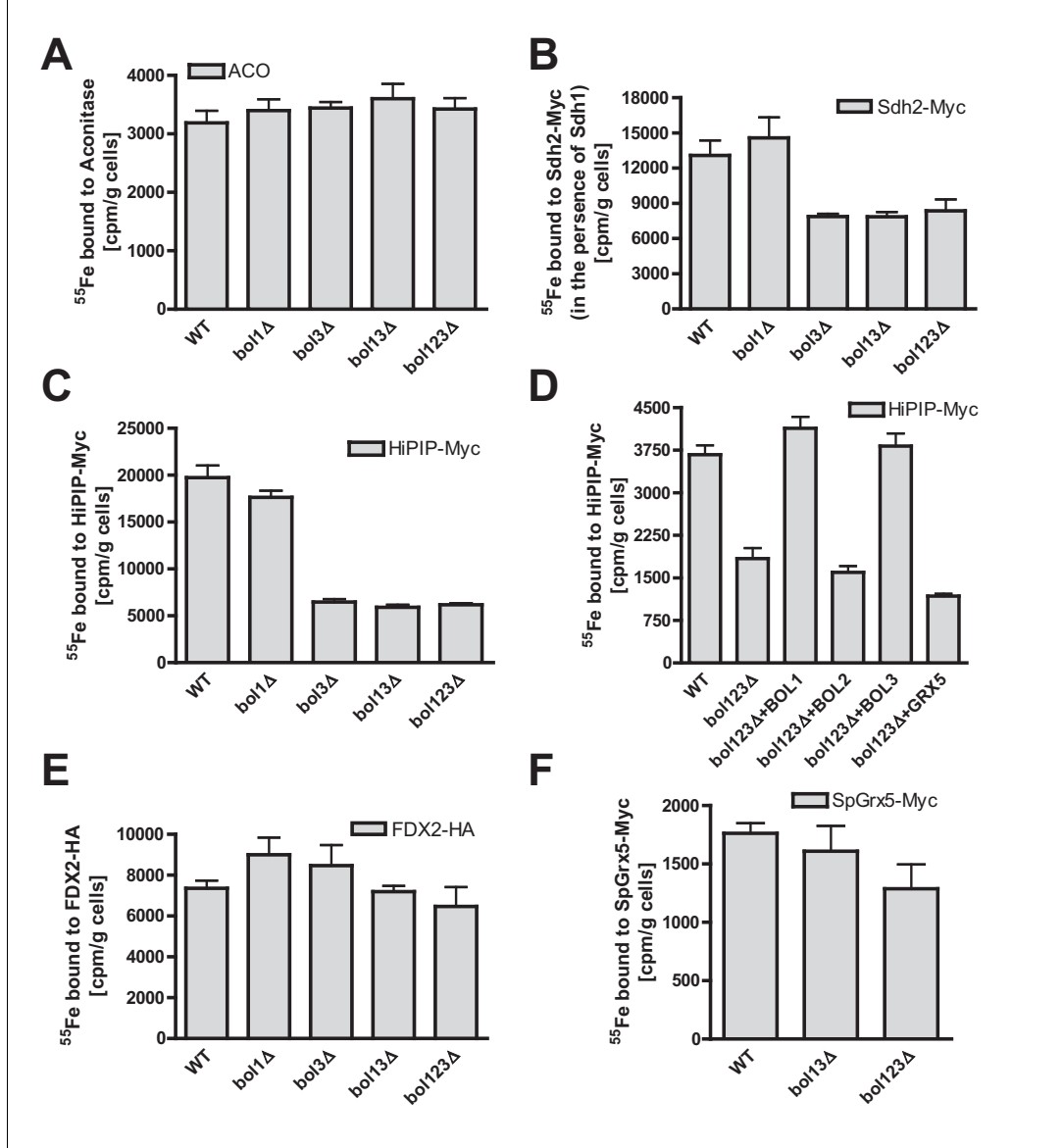

**Figure 2.** Bol1-Bol3 are required for *de novo* Fe/S cluster incorporation into specific mitochondrial [4Fe-4S] but not [2Fe-2S] proteins. (**A**–**F**) Wild-type (WT, strain BY4742) and the indicated *BOL* deletion strains were transformed with vectors overproducing (**B**) Sdh2-Myc and Sdh1, (**C** and **D**) HiPIP-Myc, (**E**) human FDX2-HA, or (**F**) *Schizosaccharomyces pombe* (Sp) Grx5-Myc. In part D, *bol123Δ* cells were additionally transformed with p414-MET25 lacking or containing *BOL1, BOL2,* or *BOL3* genes or p424-TDH3-*GRX5*. Cells were grown overnight in iron-poor SD medium and radiolabeled with 10 μCi $^{55}$Fe for 2 hr. The overproduced proteins were immunoprecipitated from cell extracts with specific antibodies. The amount of co-precipitated $^{55}$Fe was quantified by scintillation counting. Error bars indicate the SEM (n≥4).

of a subset of mitochondrial Fe/S proteins, whereby Bol1 and Bol3 perform largely overlapping, non-essential functions. Interestingly, the $^{55}$Fe radiolabeling assay revealed a slightly more important role for Bol3 as compared to Bol1. This tendency, yet less well pronounced, was also observed for the enzyme activities (see *Figure 1*). The ability of the mBols to mutually substitute each other explains why Fe/S protein defects are best visible when both *BOL1* and *BOL3* are deleted.

All Fe/S proteins tested so far contain [4Fe-4S] clusters. To get an insight into the Fe/S cluster-type specificity of mBol function, we examined the $^{55}$Fe incorporation into two mitochondrial [2Fe-2S] proteins, namely ferredoxin FDX2-HA, the human ortholog of yeast Yah1, and Grx5 from *Schizosaccharomyces pombe* (SpGrx5-Myc) (*Gelling et al., 2008*; *Uzarska et al., 2013*). The deficiency in mBols did not elicit any significant decrease in $^{55}$Fe binding to these [2Fe-2S] proteins (*Figure 2E,F*).

This behavior is strikingly different from that of members of the core ISC machinery, and shows that the mBols were not required for maturation of [2Fe-2S] proteins. We conclude that Bol1-Bol3 are dedicated ISC assembly factors specifically involved in the maturation of a subset of mitochondrial [4Fe-4S] but not of [2Fe-2S] proteins. Further, the mBols act after Grx5, the final core ISC component (*Uzarska et al., 2013*), because they were not needed for $^{55}$Fe incorporation into Grx5.

## Neither mitochondrial Bol1-Bol3 nor cytosolic Bol2 are required for cytosolic Fe/S protein biogenesis

We next examined if the mitochondrial Bol proteins perform an additional role in cytosolic Fe/S protein biogenesis, if they functionally cooperate with cytosolic Bol2, or if Bol2 alone plays a crucial role in this process. Previous studies on yeast Bol2 and human BOLA2 have suggested an involvement of these cytosolic proteins in intracellular iron metabolism (*Kumanovics et al., 2008*; *Li et al., 2012*). First, we measured the consequences of various *BOL* gene deletions on the activation of the Aft1-Aft2-dependent iron regulon, a hallmark of all cells with defects in the core ISC machinery (*Lill et al., 2012*; *Outten and Albetel, 2013*). As expected (*Kumanovics et al., 2008*), bol2Δ cells displayed a substantial activation of the Aft1-dependent *FIT3* and *FET3* genes in a GFP-based promoter assay (*Figure 3A–B*). In contrast, yeast strains lacking *BOL1* and/or *BOL3* behaved like wild-type cells indicating that the mBols are not involved in cellular iron regulation. This fits nicely to our conclusion above that Bol1-Bol3 do not belong to the core ISC system. The same situation was observed for cells lacking Nfu1 (*Figure 3B*), another known late-acting specific ISC assembly factor (*Navarro-Sastre et al., 2011*). Surprisingly, when the *BOL2* deletion was combined with genetic ablation of *BOL1* and/or *BOL3,* the iron regulon was activated slightly stronger than in bol2Δ cells (*Figure 3A–B*). This amplifying effect was specific for a Bol protein deficiency because it was reversed to the *FET3* induction level of bol2Δ cells by reintroducing *BOL1* and/or *BOL3* into bol123Δ cells, and to wild-type *FET3* levels by *BOL2* expression (*Figure 3B*).

Second, we investigated the effects of the three *BOL* gene deletions on cytosolic Fe/S proteins by assaying the enzyme activity of isopropylmalate isomerase (Leu1). No changes were observed when *BOL1-BOL3* genes were deleted either alone or in combination (*Figure 3C*). In contrast, bol2Δ cells showed a twofold decrease in Leu1 activity, which further declined upon additional deletion of *BOL1-BOL3*. Collectively, these results show that the mBols are not involved in cytosolic Fe/S protein biogenesis, yet there is a cross-talk of the mBol deficiency with the *BOL2* deletion. To better understand the reason of this combinatory effect, we took advantage of the $^{55}$Fe radiolabeling assay described above. Deletion of either *BOL1-BOL3* or *BOL2* hardly affected $^{55}$Fe/S cluster binding to Leu1 indicating that neither mitochondrial Bol1-Bol3 nor cytosolic Bol2 perform a decisive role in Leu1 Fe/S cluster incorporation (*Figure 3D,F*). However, combining the *BOL2* deletion with that of *BOL1* and/or *BOL3* resulted in an up to twofold decrease in $^{55}$Fe/S cluster association with Leu1. A similar twofold decrease (compared to wild-type) was seen for $^{55}$Fe incorporation into another cytosolic Fe/S protein, the ABC protein Rli1 using the triple *BOL* deletion strain bol123Δ (*Figure 3E–F*). We note, however, that these decreases in $^{55}$Fe incorporation were rather weak compared to defects in core ISC or CIA components (see, e.g., *Netz et al., 2007*, *2010*; *Paul et al., 2015*). The Rli1 maturation defect was completely restored when any of the *BOL* genes was reintroduced into the bol123Δ mutant. This supports the notion that neither the mBols nor Bol2 are directly involved in cytosolic Fe/S protein maturation. Obviously, their combined absence causes alterations in both mitochondrial metabolism and cellular iron homeostasis, thereby creating conditions that negatively affect cytosolic Fe/S cluster insertion. The physiological conditions prevailing in bol123Δ cells might also explain the slight decrease in the enzyme activity of aconitase which, like Leu1, is a highly susceptible protein to, e.g., oxidative stress conditions or changes in iron availability (cf. *Figures 1A* and *3C*). Together, our data indicate that the mBols execute a mitochondria-specific function in [4Fe-4S] protein assembly, whereas Bol2 plays a specific role in cellular iron regulation (*Kumanovics et al., 2008*).

## Bol1-Bol3 and Nfu1 perform distinct tasks in mitochondrial [4Fe-4S] protein maturation

Our current and previous results suggested functions of the mBols and of Nfu1 as specialized ISC assembly factors (*Cameron et al., 2011*; *Navarro-Sastre et al., 2011*). Are these functions

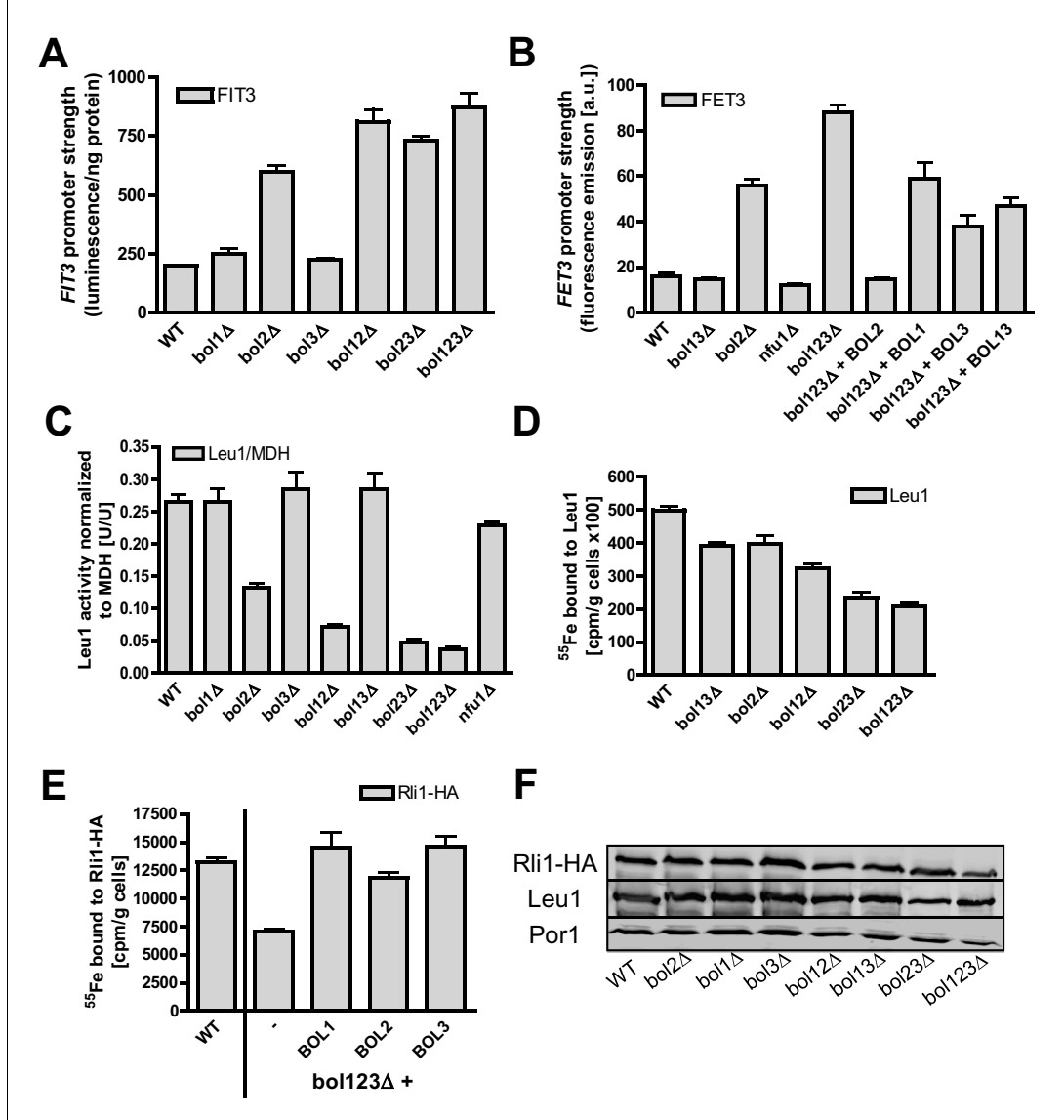

**Figure 3.** Mitochondrial Bol1-Bol3 and cytosolic Bol2 are not involved in cytosolic Fe/S protein biogenesis. Wild-type (WT, strain BY4742) and the indicated *BOL* deletion strains harboring vector (**A**) pFIT3-Luc2 or (**B**) pFET3-GFP were cultivated in iron-replete medium to mid-log phase and the activities of the *FIT3* and *FET3* promoters, respectively, were determined. In (**B**) the *bol123Δ* cells were also transformed with centromeric plasmids producing the indicated Bol proteins. Abbreviation: Bol13; Bol1 plus Bol3. (**C**) Leu1 (relative to MDH) activities were determined in the indicated deletion stains cultivated in iron-replete medium. (**D**) The indicated *BOL* deletion stains were radiolabeled with [55]Fe, and the Leu1-bound radioactivity was determined by immunoprecipitation with α-Leu1 antibodies followed by scintillation counting. (**E**) [55]Fe incorporation into Rli1-HA was determined accordingly using WT and *bol123Δ* cells transformed with centromeric plasmids producing the three Bol proteins as indicated or with the empty vector (-). (**F**) Representative immunoblots determining the protein levels of Leu1 (**D**) and Rli1-HA (**E**) in the indicated strains. Porin served as a loading control. Error bars indicate the SEM (n≥4).

independent or overlapping? We first analyzed whether simultaneous inactivation of Bol1-Bol3 and Nfu1 aggravates the Fe/S defects. As observed previously (*Navarro-Sastre et al., 2011*; *Schilke et al., 1999*), *nfu1Δ* cells showed a diminution in the activities of both aconitase and SDH relative to wild type (*Figure 4A–B*). A substantial further decrease to rather low aconitase and SDH activities was found for additional deletion of *BOL3*, but not of *BOL1*. The *BOL3* effect was not further amplified in *nfu1Δbol13Δ* cells. These findings show that Bol3 plays a distinct, Nfu1-independent

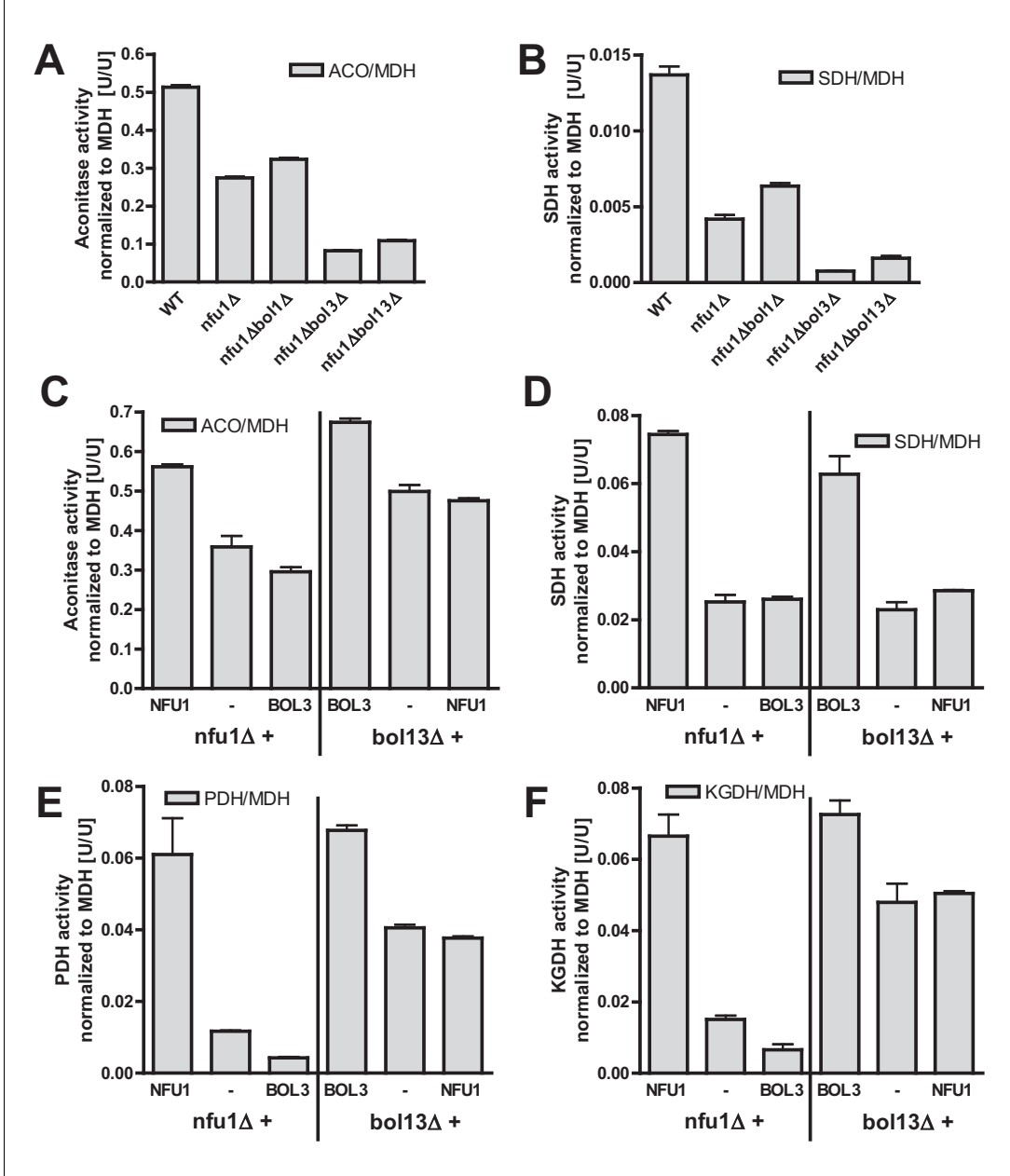

**Figure 4.** Bol1-Bol3 and Nfu1 cannot functionally replace each other in Fe/S protein maturation. (A–B) Wild-type (WT, strain BY4742) and the indicated deletion strains were grown in minimal medium containing 2% glucose and used for the preparation of mitochondria. Mitochondrial extracts were assayed for the indicated specific enzyme activities as described in *Figure 1*. (C–F) The indicated deletion strains were transformed with vector p416-MET25 lacking (empty) or containing *BOL3* or *NFU1* as indicated. Cells were grown in minimal medium containing 2% galactose and used for the preparation of mitochondria. Mitochondrial extracts were assayed for the indicated enzyme activities as described in *Figure 1*. Error bars indicate the SEM (n≥4).

The following figure supplement is available for figure 4:

**Figure supplement 1.** Growth complementation test of *nfu1Δ* and *bol13Δ* cells.

role that cannot be executed by Bol1. Apparently, even though Bol1 and Bol3 play overlapping roles (see above), their functions are not entirely identical.

We next tested whether the Bol1-Bol3 and Nfu1 proteins can mutually substitute each other's functions. To this end, we first took advantage of the growth defects of the *bol13Δ* and *nfu1Δ* strains in acetate-containing minimal medium (see above). Upon complementation of *bol13Δ* cells with *BOL1* and/or *BOL3* or of *nfu1Δ* cells with *NFU1*, wild-type growth rates were restored (*Figure 4—figure supplement 1*). In contrast, no significant growth defect complementation was observed for *nfu1Δ* cells upon *BOL1* and/or *BOL3* expression (part A). A weak complementation was seen for *bol13Δ* cells by *NFU1* expression (part B). Notably, this slight rescue effect was not observed for growth on solid acetate medium (not shown). Further, we used these cells for measuring the enzyme activities of aconitase, SDH, PDH and KGDH. In agreement with the growth behavior, Bol3 was not able to rescue any of the enzyme defects of *nfu1Δ* cells, while Nfu1 did so (*Figure 4C–F*, left). Likewise, Nfu1 failed to restore the enzyme activities of *bol13Δ* cells, while Bol3 was complementing (*Figure 4C–F*, right). These findings show that Nfu1 and Bol3 cannot mutually substitute each other's biochemical function, even upon overproduction. Despite the fact that Bol3 and Nfu1 both act late in mitochondrial Fe/S protein biogenesis, they appear to fulfil different tasks that cannot be taken over by the other protein. Hence, these data suggest that Bol1-Bol3 and Nfu1 perform individual, non-overlapping functions during Fe/S cluster assembly.

## Structural insights into the human mitochondrial BOLA proteins and the interaction with GLRX5

To extend our cell biological insights into Bol protein function, we employed NMR structural studies to examine their physical interaction with Grx5. We used the human proteins for these in vitro studies because they were more stable than the yeast counterparts. First, the solution structures of human BOLA1 and BOLA3 were determined (*Figure 5*). A detailed description of the two structures and of their backbone dynamic properties is provided in *Figure 5—figure supplement 1*. According to the DALI server (*Holm and Sander, 1993*), both BOLA1 and BOLA3 show the highest structural similarity to BolA-like proteins from *Arabidopsis thaliana*, *Babesia bovis*, and *Mus musculus* (*Abendroth et al., 2011*; *Kasai et al., 2004*; *Roret et al., 2014*). In particular, the BOLA1 structure matches better with those homologues that have a longer loop between β1 and β2, while the BOLA3 structure is more similar to homologues having a shorter β1-β2 loop (*Figure 5C* and *Figure 5—figure supplement 2*). Apparently, the structural features of the loop between β1 and β2 are key for differentiating BOLA1 from BOLA3. Remarkably, this loop is closer to the 'invariant' His residue (His102 in BOLA1 and His96 in BOLA3; *Figure 5*), which has been shown to coordinate the [2Fe-2S] cluster in the yeast Bol2-Grx3 complex (*Li et al., 2011*). This region contains further possible Fe/S cluster ligands, namely His58, His67 in BOLA1 and Cys59 in BOLA3. Another conserved His residue (His86 in BOLA1 and His81 in BOLA3) is distant from this region and thus excluded as a cluster ligand (*Figure 5*). Together, structural data indicate that BOLA1 and BOLA3 have a similar fold, but show local structural differences in the regions containing the potential Fe/S cluster ligands.

In order to characterize the interaction between BOLA1 or BOLA3 and GLRX5 at the atomic level, we first studied the apo situation. $^{15}$N labeled BOLA proteins were titrated with unlabeled apo-GLRX5, and, *vice versa*, $^{15}$N labeled apo-GLRX5 with unlabeled BOLA proteins in the presence of 5 mM GSH. Meaningful chemical shift changes were observed for both titrations in 2D $^{1}$H-$^{15}$N HSQC NMR spectra acquired upon addition of the unlabeled partner (*Figure 6—figure supplements 1A* and *2A*). The free and bound proteins are in fast and intermediate exchange regimes relative to NMR time scale; the observed chemical shift changes of the protein amide resonances saturated at a 1:1 protein ratio. $^{15}$N $R_1$ and $R_2$ NMR relaxation data performed on the final $^{15}$N labeled BOLA1-unlabelled GLRX5 or $^{15}$N labeled BOLA3-unlabelled GLRX5 mixtures showed rotational correlation times consistent with the formation of a 1:1 heterodimeric complex (*Figure 6—figure supplement 3*). NMR titrations performed by adding unlabeled apo-GLRX5 to a $^{15}$N-labeled BOLA1/$^{15}$N-labeled BOLA3 1:1 mixture showed the formation of both hetero-complexes at comparable levels, indicating similar affinities. This data fits nicely to thermophoresis results showing that both human BOLA proteins interacted with apo-GLRX5 with $K_d$ values of 3 μM (*Melber et al., 2016*; accompanying manuscript).

The chemical shift changes observed upon formation of apo-GLRX5-BOLA1 and apo-GLRX5-BOLA3 complexes were mapped on the solution structures of BOLA1, BOLA3 and apo-GLRX5. The

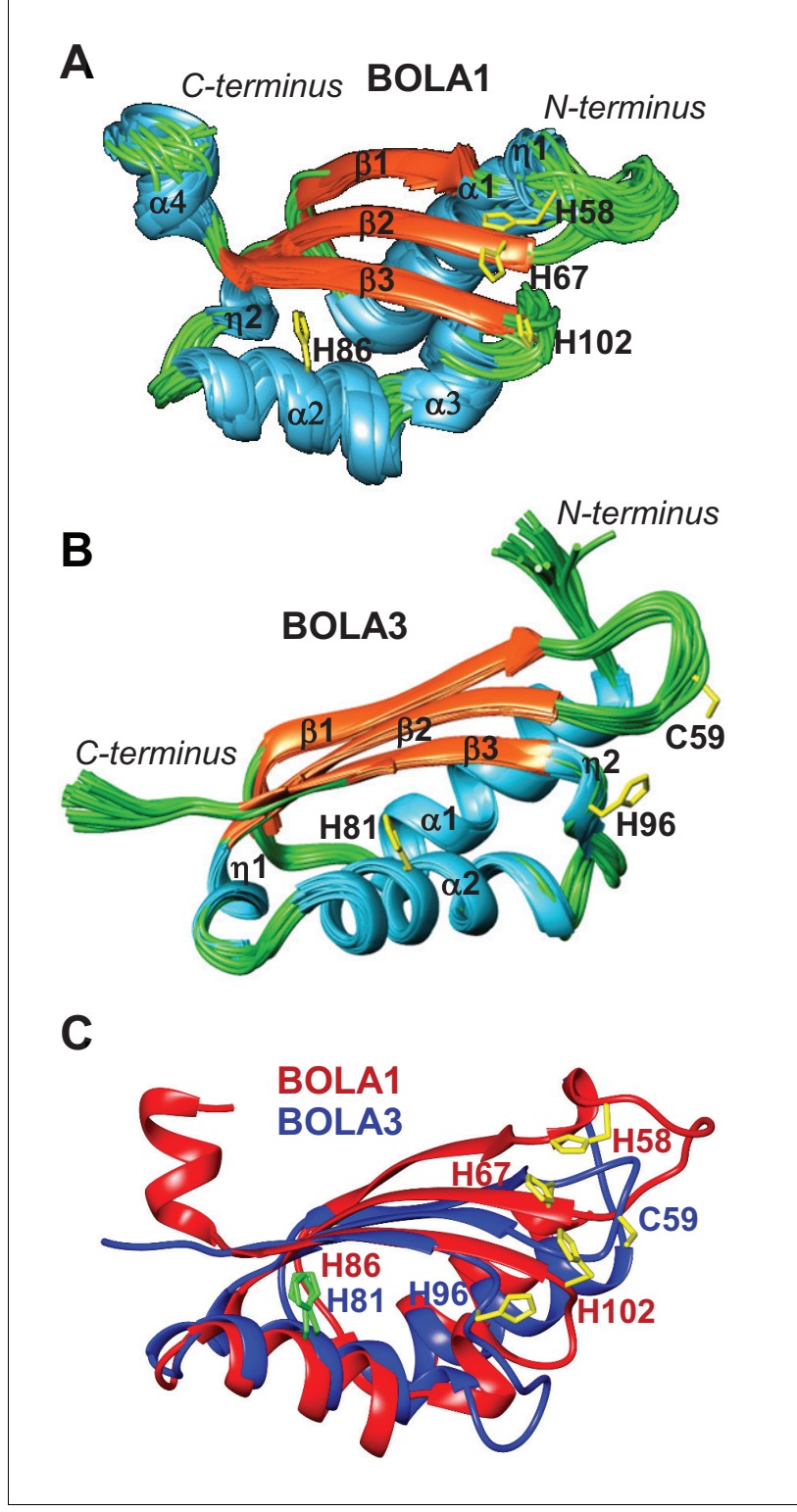

**Figure 5.** NMR solution structures of human BOLA1 and BOLA3. The structures of (**A**) BOLA1 and (**B**) BOLA3 were solved by solution NMR. Residues His58, His67 and His102 are conserved within the eukaryotic BOLA1 proteins, and residues Cys59 and His96 are conserved within the BOLA3 proteins. (**C**) Backbone superimposition of BOLA1 (red) and BOLA3 (blue) structures depicting conserved His and Cys residues in both proteins. Yellow sticks represent potential Fe/S cluster ligands, and green sticks indicate other conserved His residues.

*Figure 5 continued on next page*

*Figure 5 continued*

The following figure supplements are available for figure 5:

**Figure supplement 1.** Structural and dynamic properties of BOLA1 and BOLA3 by solution NMR.

**Figure supplement 2.** Multi-sequence alignment of the mitochondrial Bol proteins.

interaction surface on GLRX5 in the two complexes comprises the GSH binding site and its surroundings (*Figure 6A,B* and *Figure 6—figure supplement 2A*). On the BOLAs, the interactions involve helix α2, part of the β-sheet and the invariant His residue (see above). The other possible Fe/S cluster ligands (His58, His67 in BOLA1 and Cys59 in BOLA3) showed no significant chemical shifts. In conclusion, NMR data indicate that apo-GLRX5-BOLA1 and apo-GLRX5-BOLA3 complexes specifically interact involving the region surrounding the invariant His in BOLAs and the GSH binding site in GLRX5.

Recently, a heterodimeric complex formed between *A. thaliana* (At) apo-GrxS14 and AtBolA2 (showing best similarity to BOLA3 and possessing a conserved Cys in addition to the invariant C-terminal His) was characterized by NMR (*Roret et al., 2014*). Surprisingly, the human and plant complexes share only some regions of chemical shift changes, and exhibit substantial differences (*Figure 6—figure supplement 4*). For instance, the invariant His66 region of AtBolA2 does not appear to be an interaction interface, at variance with what we observed for both human BOLA complexes (*Figure 6A,B*). In addition, the GrxS14-interacting regions include the two C-terminal α4 and α5 helices, whereas helix α5 of GLRX5 of both human complexes did not show chemical shifts, and a different segment of helix α4 was involved. Moreover, helix α2 of GLRX5 showed signs of interaction only in the human BOLA-GLRX5 complexes. We conclude that there is a high degree of organismic variability in the precise interaction mode between Grx and Bol proteins.

We next investigated the interaction of holo-GLRX5 with the BOLA proteins by chemically reconstituting a [2Fe-2S] cluster on equimolar GLRX5 and BOLAs. NMR $^{1}$H-$^{15}$N HSQC spectra were acquired for the co-reconstituted complexes of $^{15}$N labeled BOLA3 or BOLA1 with GLRX5. The $^{1}$H-$^{15}$N HSQC spectra of these co-reconstituted complexes were similar to spectra obtained by mixing BOLA1 or BOLA3 with [2Fe-2S]-containing holo-GLRX5 (not shown). Upon overlaying the $^{1}$H-$^{15}$N HSQC spectra of the holo-complexes with those acquired for the corresponding apo-complexes, meaningful spectral variations (either as chemical shift changes and/or line-broadenings beyond signal detection) were detected on both proteins (*Figure 6—figure supplements 1B* and *2B*). The $^{1}$H-$^{15}$N HSQC maps of the co-reconstituted complexes having $^{15}$N labeled GLRX5 with unlabeled BOLA1 or BOLA3 were not superimposable to the $^{1}$H-$^{15}$N HSQC map of the holo-GLRX5 homodimer indicating formation of [2Fe-2S] hetero-complexes $^{15}$N $R_1$ and $R_2$ NMR relaxation data of BOLA1 or BOLA3, in the [2Fe-2S] chemically reconstituted complexes with GLRX5, indicated no significant changes in protein size with respect to the apo-complexes. Overall, the NMR data showed the presence of 1:1 heterodimeric holo-complexes between GLRX5 and BOLA1 or BOLA3.

Chemical shift perturbation and line broadening analyses were performed by comparing the $^{1}$H-$^{15}$N HSQC spectrum of chemically reconstituted $^{15}$N-labeled BOLA3-GLRX5 with that of BOLA3. The residues of the loop containing the conserved Cys59 of BOLA3 showed strong chemical shift changes only in the holo hetero-complex relative to the apoform (*Figure 6B,D* and *Figure 6—figure supplement 1*, right parts). A similar comparison for the $^{15}$N-labeled BOLA1-GLRX5 complex showed that the region around His67 of BOLA1 (structurally close to Cys59 of BOLA3) exhibited similar chemical shift changes, even though fewer residues were affected in BOLA1 (*Figure 6A,C*, and *Figure 6—figure supplement 1*, left parts). The chemical shift perturbation analysis for $^{15}$N-labeled GLRX5 revealed that the residues of GLRX5 involved in the apo interaction are also affected in the holo-complex and that the conserved Cys67 of GLRX5 was additionally altered in the holo complexes only (*Figure 6—figure supplement 2*). In addition, two other adjacent GLRX5 residues (Lys59 for BOLA3 and Gly60 for BOLA1 interaction) were altered that are part of the GSH binding site. The regions of GLRX5 affected by the interaction with BOLA1 and BOLA3 are essentially the same as those in the [2Fe-2S]$^{2+}$ GLRX5 homodimer (*Banci et al., 2014*). This indicates that, for GLRX5, the

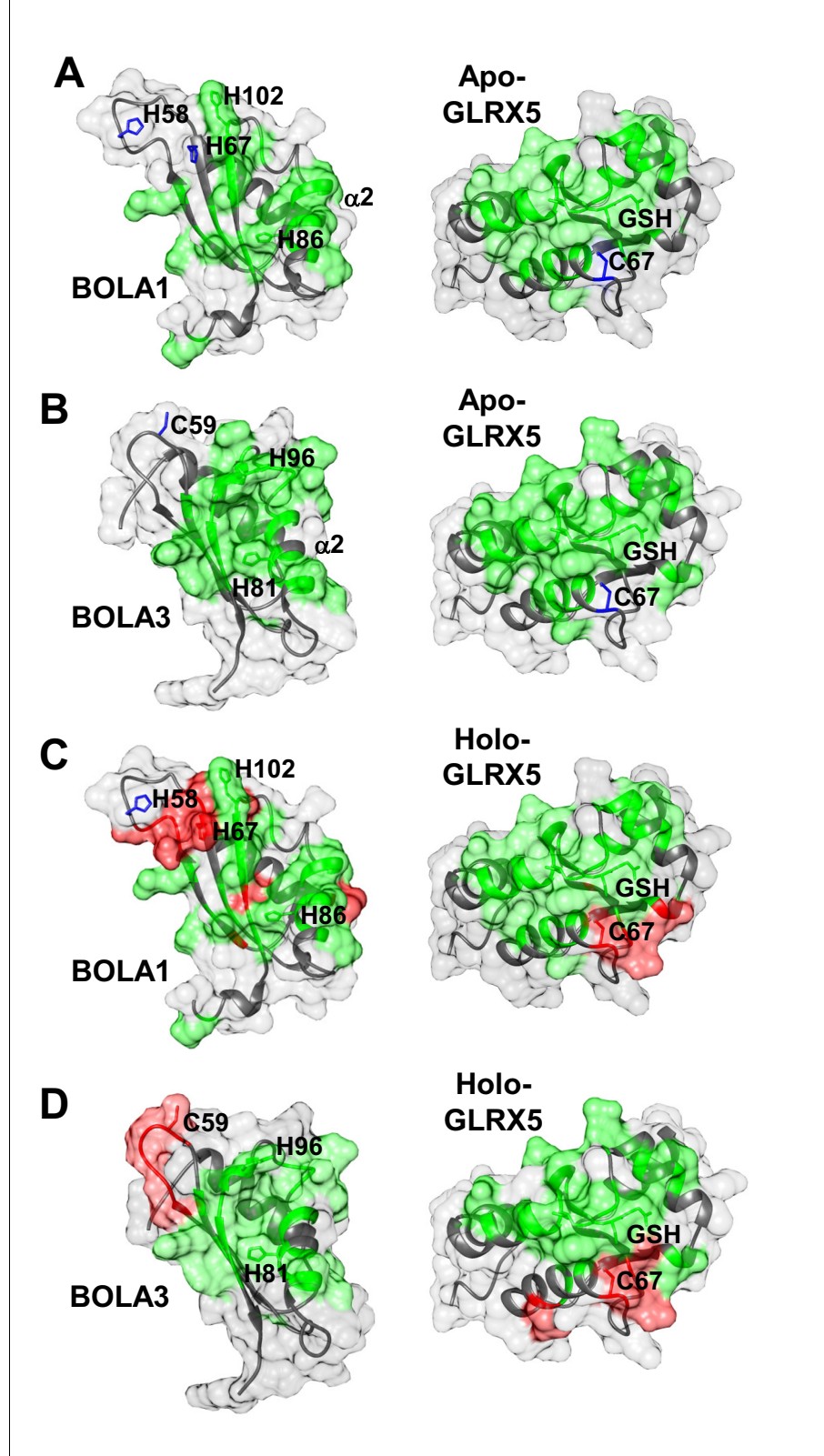

**Figure 6.** Structural basis of the interaction between human apo- and holo-GLRX5 with the BOLA proteins. (**A** and **B**) Backbone chemical shift differences, obtained from a comparison of the $^1$H-$^{15}$N HSQC spectra of apo-GLRX5 or BOLA proteins with that of (**A**) BOLA1-apo-GLRX5 or (**B**) BOLA3-apo-GLRX5 (1:1 mixture in buffer N), were mapped on the solution structures of the proteins. (**C** and **D**) Backbone chemical shift differences, obtained from a

*Figure 6 continued on next page*

*Figure 6 continued*

comparison of the $^1$H-$^{15}$N HSQC spectrum of apo-GLXR5 or BOLA proteins with that of (**C**) BOLA1-apo-GLRX5 or (**D**) BOLA3-apo-GLRX5 (1:1 mixture in buffer N) chemically reconstituted with a [2Fe-2S] cluster, were mapped on the solution structures of the proteins. Green regions show residues with significant chemical shift changes (that is both in terms of chemical shift and broadening beyond detection effects) observed upon formation of the apo- or holo-complexes. Red areas depict those residues additionally affected upon holo-complex formation. Critical residues and GLRX5-bound GSH are depicted as sticks.

The following figure supplements are available for figure 6:

**Figure supplement 1.** Chemical shift changes upon complex formation between GLRX5 and BOLA proteins monitoring backbone chemical shift changes of BOLA1 and BOLA3.

**Figure supplement 2.** Chemical shift changes upon complex formation between GLRX5 and BOLA proteins, monitoring backbone chemical shift changes of GLRX5.

**Figure supplement 3.** Experimental and predicted rotational correlation times ($\tau_C$).

**Figure supplement 4.** Comparison of the apo-AtGrxS14-AtBolA2 complex with apo-GLRX5/BOLA3.

---

same interaction regions are involved in both homo- and hetero-dimer formation, and the same groups (the conserved Cys67 and GSH) can act as iron ligands of the [2Fe-2S]$^{2+}$ cluster.

## Preferential interaction of BOLA1 with holo-GLRX5 and of BOLA3 with holo-NFU1

We used circular dichroism (CD) spectroscopy to follow the characteristics of Fe/S cluster coordination upon ligand exchange from the holo-Grx5 homo-dimer to the Grx5-Bol hetero-dimer (*Banci et al., 2015*; *Li et al., 2009*; *Li et al., 2012*). GLRX5 and stoichiometric amounts of BOLA1 or BOLA3 were chemically co-reconstituted with a [2Fe-2S] cluster and analyzed by CD (*Figure 7*). Similar results were obtained when GLRX5 was reconstituted first and then the BOLA proteins were added (*Figure 7—figure supplement 1*). The CD spectrum of GLRX5-BOLA1 characteristically differed from that of holo-GLRX5 in a positive versus negative ellipticity around 400 nm with a 6 nm blue shift and 40% decrease of the ~460 nm peak with a 6 nm red shift (*Figure 7A*). These spectral changes saturated at an equimolar ratio of holo-GLRX5 and BOLA1 suggesting a high affinity equimolar hetero-complex formation with a shared [2Fe-2S] cluster (*Figure 7—figure supplement 1B*). The cluster coordinated by the GLRX5-BOLA1 hetero-dimer was stable against reduction by dithionite creating a CD spectrum in which the peak at 460 nm was maintained but that at 400 nm was lost. In contrast, the cluster of holo-GLRX5 was destroyed upon dithionite treatment with no residual CD peaks (*Figure 7A* and *Figure 7—figure supplement 1A*). These results demonstrate that BOLA1 stabilized the shared [2Fe-2S] cluster in the hetero-dimer with GLRX5.

A strikingly different Fe/S cluster behavior was observed upon BOLA3-GLRX5 complex formation. Hardly any CD spectral changes were observed upon co-reconstituting or mixing GLRX5 and BOLA3 (*Figure 7B* and *Figure 7—figure supplement 1C,D*). Further, the Fe/S cluster was not protected against destruction by dithionite treatment indicating rather different coordination situations for the [2Fe-2S] clusters in the two GLRX5-BOLA heterodimers. These results are consistent with the different NMR chemical shift changes in the putative Fe/S cluster binding regions of the two BOLAs in the hetero-complexes (*Figure 6C,D*). They do not explain, however, why NMR still detected holo-specific changes in the putative cluster-binding region of the BOLA3 hetero-complex (*Figure 6D*). Since the NMR data were recorded in GSH-containing buffer at lower pH, we repeated the CD analysis under NMR buffer conditions. We found a general decrease of the CD peaks upon GLRX5-BOLA3 hetero-complex formation (*Figure 7C*). These changes mainly depended on the presence of GSH, and not salt or pH (*Figure 7D*). In all cases, the CD signal decrease did not reach saturation even after addition of a more than tenfold molar excess of BOLA3 over holo-GLRX5. This result indicated a rather low affinity for this interference of BOLA3 with the [2Fe-2S] cluster. Overall, the CD data

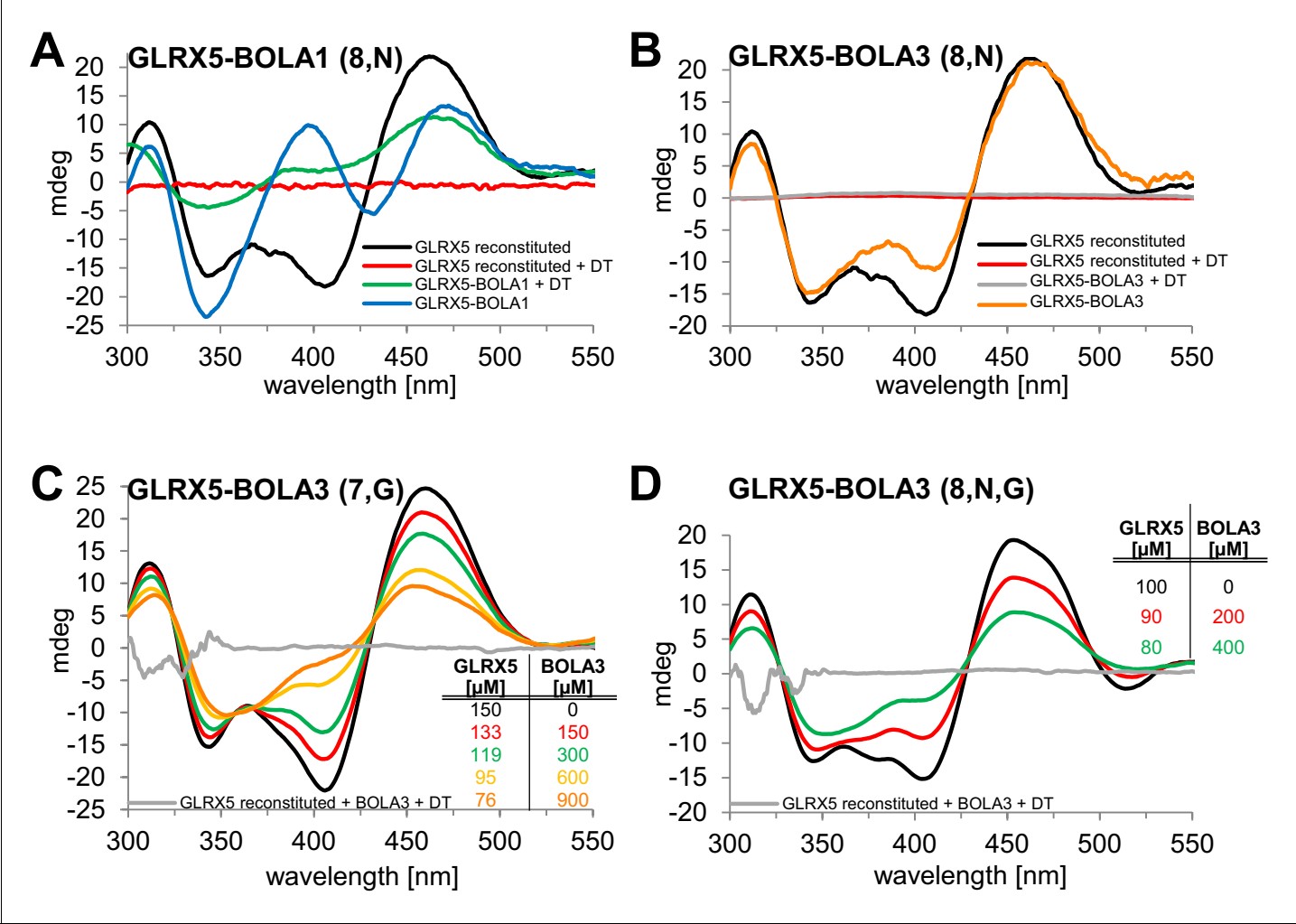

**Figure 7.** Human BOLA1 but not BOLA3 stabilizes the [2Fe-2S] cluster of holo-GLRX5 upon heterodimer formation. (**A–B**) Chemical reconstitution of Fe/S clusters in buffer R was performed with GLRX5 in the absence or presence of stoichiometric amounts of (**A**) BOLA1 or (**B**) BOLA3, and CD spectra were monitored under anaerobic conditions. Additional spectra were recorded after addition of 2 mM dithionite (DT). (**C–D**) Chemically reconstituted GLRX5 was titrated with the indicated concentrations of BOLA3 in (**C**) buffer N or (**D**) buffer P, and CD spectra (corrected for dilution) were recorded under anaerobic conditions. Abbreviations for varied buffer conditions: 7, 8: pH; N, 150 mM NaCl; G, 5 mM GSH.
The following figure supplements are available for figure 7:

**Figure supplement 1.** Stoichiometric hetero-complex formation of human holo-GLRX5 with BOLA1 results in characteristic CD spectral changes indicating shared binding of the [2Fe-2S] cluster.

**Figure supplement 2.** Gel filtration verifies the complex formation between human holo-GLRX5 and BOLA1.

**Figure supplement 3.** Preferential binding of human BOLA3 to the holo-form of NFU1.

indicate a rather weak and GSH-dependent influence of BOLA3 on the GLRX5 [2Fe-2S] cluster which is not stabilized by hetero-complex formation.

To independently analyze the GLRX5-BOLA complex formation, we used size exclusion chromatography. Holo-GLRX5 (recorded at 420 nm) eluted at 44 kDa indicative of homodimer with bound Fe/S cluster (*Figure 7—figure supplement 2A*). A significantly smaller molecular mass was observed for the heterodimeric GLRX5-BOLA1 (32 kDa), while the mixture of holo-GLRX5 and BOLA3 did not alter the elution behavior of GLRX5 at 420 nm. The elution profile at 280 nm showed no significant

protein peak around 30 kDa that might represent a GLRX5-BOLA3 heterodimer (*Figure 7—figure supplement 2B*). Further, BOLA3 from the GLRX5-BOLA3 mixture eluted exclusively at the position of BOLA3 alone. These results support the formation of a stable heterodimer between GLRX5 and BOLA1, yet the interaction of GLRX5 with BOLA3 appears to be kinetically unstable, especially in the absence of GSH.

The lack of stable interaction between GLRX5 and BOLA3 raised the question of whether the latter could interact with NFU1. Since Fe/S cluster binding to NFU1 was CD-silent, we performed in vitro binding studies by microscale thermophoresis (MST), an equilibrium method that can be performed under anaerobic conditions (*Webert et al., 2014*). For these interaction experiments we used both apo-NFU1 and chemically reconstituted holo-NFU1 carrying a [4Fe-4S] cluster (*Tong et al., 2003*). MST indeed showed a specific interaction between the two BOLA proteins and both apo- and holo-NFU1 (*Figure 7—figure supplement 3*). The dissociation constants ($K_d$) of the interactions were in the range of 3 μM, with the notable exception of a fourfold higher affinity detected for BOLA3 and holo-NFU1 ($K_d$ = 0.8 μM). As a control, no significant affinity was observed for human [2Fe-2S] ferredoxin FDX2, cytochrome *c* or RNase A as control proteins. Collectively, these results data suggest that holo-GLRX5 preferentially cooperates with BOLA1, and holo-NFU1 with BOLA3.

## Discussion

In this work, we have defined the molecular function of the mitochondrial Bol1 and Bol3 proteins (mBols) as specific ISC assembly factors required for the insertion of [4Fe-4S] clusters into a small subset of mitochondrial target apoproteins. The two mBols act late in the ISC pathway, and execute a largely overlapping function, because only simultaneous deletion of both *BOL* genes was associated with appreciable effects on mitochondrial [4Fe-4S] target proteins (*Figure 8*). Lipoic acid synthase (LIAS) with its two [4Fe-4S] clusters is the primary client of mBol function because of strong effects on lipoic acid-dependent proteins (such as PDH and KGDH) in *bol13Δ* cells. These severe effects were observed under all experimental conditions tested, unlike the comparatively weak effects on SDH (complex II) which typically is one of the more sensitive mitochondrial Fe/S proteins upon ISC factor defects. Therefore, we cannot exclude an indirect effect of the *BOL* gene deletion on SDH maturation. The [4Fe-4S] aconitase, on the other hand, was not or hardly affected in *bol13Δ* cells distinguishing the *BOL* deletion phenotype from that of all other known ISC genes, with the exception of the complex I-specific *IND1* (*Bych et al., 2008*; *Sheftel et al., 2009*). We propose that the rather inconspicuous effects on aconitase are indirect consequences of the metabolic and respiratory alterations arising from lipoic acid-dependent enzyme defects which mainly affect the citric acid cycle (PDH and KGDH) and the amino acid metabolism (BCKDH, GCS; *Figure 8*). The functions of the mBols are largely overlapping, yet not entirely identical. This was evident from small effects on LIAS and SDH in *bol3Δ* cells, versus no detectable alterations upon *BOL1* deletion. Moreover, double deletion of *BOL3* and *NFU1*, encoding another late-acting ISC factor (*Navarro-Sastre et al., 2011*), (*Melber et al., 2016*; accompanying manuscript), exacerbated the Fe/S protein defects compared to single deletions, while double deletion of *BOL1* and *NFU1* behaved similarly to *nfu1Δ* cells (*Figure 4A,B*). Two models may explain the largely complementary functions of the two mBols. i) The mBols assist the same biochemical reaction. In this case, the subtle differences between *bol1Δ* and *bol3Δ* strains may be due to minor target apoprotein specificities of the two mBols or the different protein interactions with Grx5 and Nfu1. ii) The mBols act consecutively in two independent maturation steps that both can be bypassed to some extent. Even though we biochemically favor the first possibility, current knowledge does not allow a more precise mechanistic definition of mBol function. Taken together, the major evident physiological function of the yeast mBol proteins is in lipoate synthase maturation. A similar defect is seen in human cells obtained from *BOLA3* patients (*Baker et al., 2014*; *Cameron et al., 2011*; *Haack et al., 2013*). This phenotypical resemblance suggests that the yeast mBols and at least human BOLA3 are functionally similar. Even though yeast appears to be a suitable model system for the physiological and mechanistic investigation of mBols, the two human BOLAs now need to be studied in cell culture to gain better insights into their relative function.

Several findings allowed us to define the site of action of the mBols in the multi-step ISC assembly pathway (*Figure 8*). First, the mBols were not required for maturation of mitochondrial [2Fe-2S]

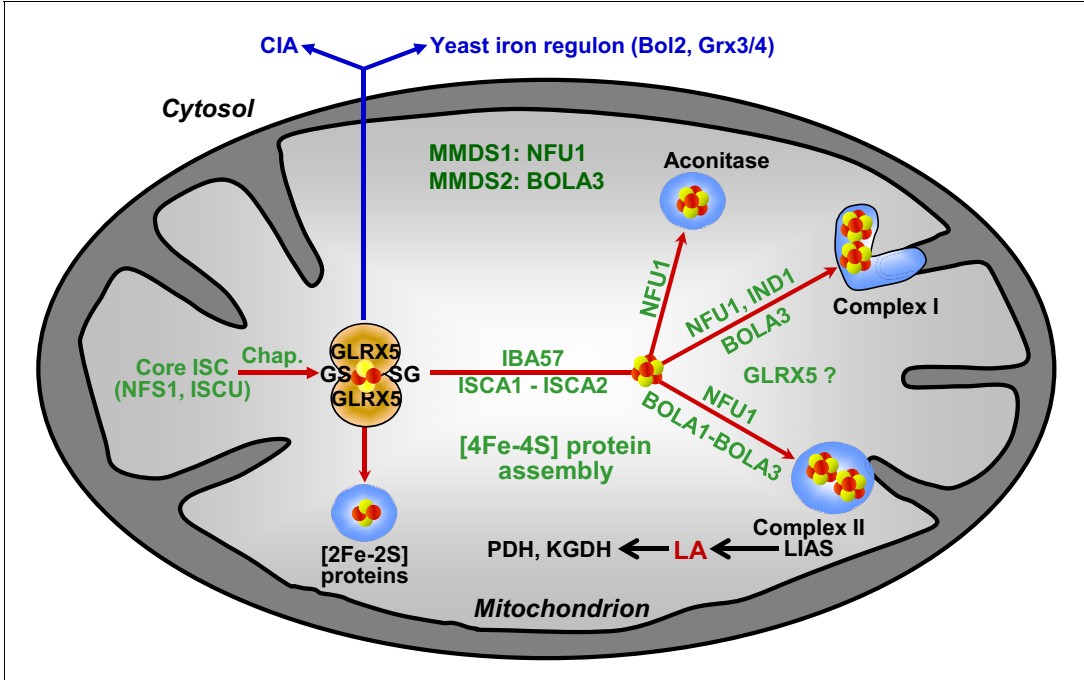

**Figure 8.** Working model for the role of mitochondrial BOLA proteins as specific ISC assembly factors in the late phase of mitochondrial Fe/S protein biogenesis. Members of the core ISC assembly machinery including the sulfur donor NFS1, the scaffold protein ISCU, and dedicated chaperones (Chap.) mediate the assembly of a transiently bound, glutathione (GS)-coordinated [2Fe-2S] cluster on the monothiol glutaredoxin GLRX5 which is essential for [2Fe-2S] protein maturation, cytosolic Fe/S protein assembly (CIA), and cellular iron regulation in yeast. With the help of ISCA1, ISCA2, and IBA57, the GLRX5-bound [2Fe-2S] cluster is converted into a [4Fe-4S] type. BOLA1 and BOLA3 together with NFU1 and IND1 function in a specific assembly of mitochondrial [4Fe-4S] proteins as indicated. The central target of the BOLA proteins is the Fe/S protein lipoic acid (LA) synthase (LIAS). Its product is used as a cofactor of five mitochondrial enzymes including pyruvate dehydrogenase (PDH) and 2-ketoglutarate dehydrogenase (KGDH). Based on the hetero-complex formation with the BOLA proteins an additional function of GLRX5 in this late phase of mitochondrial Fe/S protein assembly is likely. Mutations in human *NFU1* and *BOLA3* cause multiple mitochondrial dysfunction syndromes (MMDS; for review see [*Beilschmidt and Puccio, 2014*; *Stehling et al., 2014*]).

proteins or cytosolic Fe/S proteins. Second, their deletion did not affect the yeast cellular iron regulon. These criteria distinguish the mBol phenotype from that of the components of the core ISC machinery including Grx5, that is the final core ISC factor (*Uzarska et al., 2013*), and suggest that mBols act subsequently to Grx5. This view is further corroborated by our finding that, third, [55]Fe incorporation into Grx5 occurred independently of mBols. Finally, the function of the mBols must be after the involvement of Isa-Iba57 proteins in the [4Fe-4S] cluster formation, because the mBols exhibit high [4Fe-4S] target specificity. In Isa-Iba57-defective cells, all mitochondrial [4Fe-4S] proteins and the non-Fe/S protein cytochrome oxidase are severely affected, unlike in *bol13Δ* cells (*Mühlenhoff et al., 2011*; *Sheftel et al., 2012*). Overall, the site of action of the mBols is similar to that of the late-acting ISC assembly factor Nfu1 (*Navarro-Sastre et al., 2011*) (*Figure 8*). However, we found that the mBol and Nfu1 proteins cannot complement each other's function, even after overexpression, suggesting that they both fulfil a highly specific task in the dedicated insertion of [4Fe-4S] clusters into target apoproteins.

The observation that the mBols function subsequently to Grx5 is somewhat unexpected in light of our findings that the mBols form binary complexes with this core ISC component. Apparently, complex formation with the mBols is not critical for Grx5's major function as a core ISC factor (*Figure 8*), as the *GRX5* deficiency is associated with rather severe defects in virtually all mitochondrial and cytosolic Fe/S proteins plus a misregulation of cellular iron homeostasis (*Rodriguez-Manzaneque et al., 2002*; *Uzarska et al., 2013*; *Ye et al., 2010*), radically distinguishing this phenotype from that of the rather inconspicuous *BOL1-BOL3* deletion. It is well possible that the Grx5-mBol complexes play a role at the later stage of Fe/S cluster insertion into apoproteins. This seems likely based on

interaction between BOLA1 and GLRX5 observed in human cells (*Willems et al., 2013*), and similar interactions reported in the accompanying manuscript (*Melber et al., 2016*). At present, it is impossible to verify such a function by in vivo studies, because the potential Grx5 involvement at this later stage of the ISC pathway is hidden by the severe *GRX5* deletion phenotype.

In our present work, we have characterized the Grx5-mBol interactions by a number of in vitro techniques including gel filtration, CD spectroscopy and NMR structural analyses. Human GLRX5 interacted with both mBols, stoichiometrically forming a 1:1 complex both in the apo- and chemically reconstituted holo-forms, involving similar regions on the three proteins as revealed by NMR studies. However, for the holo-forms, further residues were affected, and were clustered around the region containing the invariant histidine (His102 in BOLA1 and His96 in BOLA3; *Figure 7C,D*), which in cytosolic Bol2 is a ligand for Fe/S cluster binding with Grx3-Grx4 (*Li et al., 2011*). These Bol regions include other potential cluster ligands, which are two His residues in BOLA1, located at the end of β2 and in the loop between β1 and β2, and one Cys in BOLA3, located in the loop between β1 and β2. CD spectroscopy revealed striking differences in Fe/S cluster binding to the two holo-GLRX5-BOLA complexes. Compared to holo-GLRX5, the holo-GLRX5-BOLA1 complex showed characteristic new spectral features including a new peak at ~400 nm. GLRX5 and BOLA1 coordinate the [2Fe-2S] cluster with high affinity. The spectral features of the complex were rather insensitive to changes in ionic conditions or to the presence of additional GSH. Most characteristically, the [2Fe-2S] cluster of GLRX5 was stabilized by BOLA1 binding against destruction by the reductant dithionite indicating a stable holo-GLRX5-BOLA1 complex.

All these criteria were different for the holo-GLRX5-BOLA3 complex. First, the CD spectral features of the holo-GLRX5-BOLA3 complex differed from those of holo-GLRX5 by a general signal decrease, and were observed mainly in the presence of GSH suggesting that without added GSH the GLRX5 cluster is not shared with BOLA3. Second, the spectral changes induced by BOLA3 addition to holo-GLRX5 were not saturable indicating a rather weak influence of BOLA3 on the GLRX5 Fe/S cluster. Third, the CD signals became rather weak at high BOLA3 or salt concentrations. Finally, dithionite readily destroyed the [2Fe-2S] cluster of GLRX5 in the presence and absence of BOLA3. In support of this data, gel filtration did not identify a stable complex between holo-GLRX5 and BOLA3 indicating that their interaction is kinetically labile, and a hetero-dimer may dissociate during the non-equilibrium method. All these findings document radically different Fe/S cluster binding properties of the two GLXR5-BOLA complexes with BOLA1 stabilizing and BOLA3 destabilizing the [2Fe-2S] cluster. Interestingly, we found a preferential interaction of BOLA3 with the holoform of NFU1 by thermophoresis. These data suggest that BOLA3 may stabilize the [4Fe-4S] cluster bound on NFU1. A specific genetic interaction of *NFU1* and *BOL3* supports our biochemical findings. Double deletion of these genes exacerbated the defects of SDH and LIAS, whereas simultaneous deletions of *NFU1* and *BOL1* did not (*Figure 4A and B*). This suggests that Bol3 and Nfu1 cooperate in SDH and LIAS maturation. Future biochemical studies have to address the physiological meaning of the stabilizing or destabilizing function of the mBols for the different Fe/S clusters on Grx5 and Nfu1.

The NMR structures of the mBols share a similar fold, alike to those of other eukaryotic homologous (*Abendroth et al., 2011*; *Kasai et al., 2004*; *Roret et al., 2014*). Further, similar interacting regions for the apo- and holo-states of the GLRX5-BOLA heterodimers were found, and a similar location of the invariant His residue, a known Fe/S cluster ligand in Bol2 (*Li et al., 2011*). However, we note that its surrounding is structurally different in the two BOLAs, that is His102 of BOLA1 is the first residue following a long β-strand, while His96 of BOLA3 is in a $3_{10}$ helix (*Figure 6C*). The other potential Fe/S cluster ligands are located in structurally different positions of the two mBols. In particular, the location of the two conserved His58, His67 residues in BOLA1 and of Cys59 in BOLA3 are different (*Figure 6C*). These features can possibly explain the observed differences in the binding and stability of [2Fe-2S] clusters to the holo heterodimers with GLRX5. The distinct functional roles of the mBols could be viewed as consequence of the different coordination spheres of the Fe/S cluster.

The role of the mBols in a late step of the mitochondrial ISC pathway fits well with our observation that their functional deficiency (*bol13Δ* cells) does not cause any significant defects on cytosolic Fe/S proteins. Interestingly, a combination of these gene deletions with *BOL2* elicited significant effects, even though these were rather weak compared to known core ISC or CIA deficiencies (see, e.g., *Gerber et al., 2004*; *Hausmann et al., 2005*; *Netz et al., 2010*). The defects, especially in the

triple deletion mutant *bol123Δ*, are therefore best explained by combined detrimental effects of mitochondrial respiratory and metabolic changes arising from the *BOL1-BOL3* deletion (see above) and the disturbed cellular iron homeostasis resulting from *BOL2* ablation. Single *BOL2* deletion is not associated with major phenotypes apart from iron homeostasis (*Kumanovics et al., 2008*) (*Figure 1—figure supplement 3*). This is clearly different from the usual lethality of CIA gene deletions (*Netz et al., 2014*; *Paul and Lill, 2015*), and fits well with the rather weak or even inconspicuous effects of a *BOL2* deletion on the maturation of the cytosolic Fe/S proteins Leu1 and Rli1. Any slight effects are most likely due to the auxiliary role of Bol2 in iron metabolism (*Kumanovics et al., 2008*). In this function, Bol2 cooperates with Grx3-Grx4 (human PICOT) which, in its role in cellular iron mobilization, is essential for cytosolic Fe/S protein maturation (*Banci et al., 2015*; *Haunhorst et al., 2013*; *Mühlenhoff et al., 2010*).

Our comprehensive physiological investigations of mBol function in yeast reveal that *bol13Δ* cells behave similarly to human patients deficient in BOLA3 (*Baker et al., 2014*; *Cameron et al., 2011*; *Haack et al., 2013*). Patients with the resulting multiple mitochondrial dysfunction syndrome 2 (MMDS 2) show major defects in lipoic acid-dependent enzymes, similar to the most severe effect observed in *bol13Δ* yeast. Moreover, the patient cells are deficient in respiratory complexes I and II, the latter also being affected in yeast. We conclude that the mBol protein function is conserved from yeast to man with a major role in [4Fe-4S] cluster insertion into LIAS and respiratory complexes I and II, whereas Fe/S proteins such as aconitase are matured independently of mBol function. The conclusion of similar mBol functions in yeast and man may be surprising based on the rather inconspicuous yeast deletion phenotypes versus the lethal effect in BOLA3-deficient individuals. This difference may simply reflect the higher complexity of a multi-cellular organism and its dependence on respiration and metabolic homeostasis, whereas yeast can tolerate substantial deviations from optimal conditions. Notably, RNAi-mediated depletion of BOLA1 was associated with rather mild effects (*Willems et al., 2013*) predicting that the combination with a BOLA3 depletion may create a more pathological phenotype mimicking that of human patients and of *bol13Δ* yeast cells. Our elucidation of the target apoprotein-specific ISC assembly function of mBols will be instrumental for future mechanistic studies on how the mBols facilitate the insertion of [4Fe-4S] clusters into LIAS and possibly other Fe/S proteins. These studies will also have to reveal why other Fe/S proteins such as aconitase are matured efficiently in the absence of these specialized ISC assembly factors.

## Materials and methods

### Yeast strains and growth conditions

*S. cerevisiae* strains used in this study are listed in *Supplementary file 1A*. Cells were cultivated in rich medium (YP) or synthetic minimal medium (SC) supplemented with amino acids as required and 2% (w/v) glucose, galactose or acetate, or 3% (w/v) glycerol (*Sherman, 2002*). Iron-depleted minimal media were prepared using yeast nitrogen base without $FeCl_3$ (ForMedium). Plasmids used in this study are listed in *Supplementary file 1B*. Plasmid constructs were verified by DNA sequencing and/or functional complementation of a corresponding yeast mutant.

### Purification of recombinant proteins

Genes of human BOLA1 (residues 30–137), BOLA3 (25–107), GLRX5 (32–157) and NFU1 (57–254) (each lacking its mitochondrial presequence) were inserted into the multiple cloning site I of pET-Duet1 vector (Novagen-Merck, Darmstadt, Germany) with a N-terminal His-tag. N-terminally tagged proteins were purified from *E. coli* BL21 (DE3) using a HisTrap HP column (GE Healthcare). The eluted proteins were treated with 5 mM DTT and isolated by gel filtration on a Superdex 200 16/60 gel filtration column (GE Healthcare) in reconstitution buffer R (50 mM Tris-HCl, pH 8.0, 150 mM NaCl, 5% glycerol). For NMR the BOLA proteins (BOLA1, 21–137; BOLA3, 27–107) were obtained from BL21(DE3)GOLD *Escherichia coli* (Novagen) transformed with pETG20A vector (BOLA1) and pET15 vector (BOLA3). Cells were grown at 37°C in LB or minimal media with ($^{15}NH_4)_2SO_4$ and/or $^{13}C$-glucose containing 100 µg/mL ampicillin. Both proteins contained a N-terminal His-tag, which was cleaved by tobacco etch virus protease in 50 mM Tris-HCl pH 8, 0.5 mM EDTA, 5 mM GSH, and 1 mM DTT overnight at room temperature. GLRX5 protein in its apo- and holo-forms was produced as previously reported (*Banci et al., 2014*).

## Chemical reconstitution of Fe/S proteins

All solutions used for chemical reconstitution were prepared freshly in a COY anaerobic chamber by dissolving in degassed water. Protein samples were reduced in buffer R containing 5 mM DTT for 2–3 hr on ice in buffer R containing a two fold excess of GSH. Reconstitution was initiated at room temperature by the addition of a 2–5-fold excess of ferric ammonium citrate. After five minutes a 2–5-fold excess of lithium sulfide was slowly added. After 2 hr reconstituted proteins were desalted on a PD-10 column (GE Healthcare) equilibrated with buffer R. Incorporation of the Fe/S clusters into apoproteins was verified by UV-Vis (V-550, Jasco Inc.) and CD spectroscopy (J-815, Jasco Inc.) and the determination of bound iron and acid-labile sulfur (*Pierik et al., 1992*).

## Circular dichroism spectroscopy

CD spectroscopy (J-815, Jasco Inc.) of human GLRX5 reconstituted either alone (>85% holo based on iron and acid-labile sulfur determination) or in the presence of stoichiometric amounts of either BOLA1 or BOLA3 was performed anaerobically (protein concentrations 200–300 μM) in buffer R, buffer N (50 mM phosphate buffer pH 7, 5 mM GSH, and 5 mM DTT), or buffer P (50 mM phosphate buffer pH 8, 150 mM NaCl, 5 mM GSH). Alternatively, reconstituted holo-GLRX5 was titrated with increasing concentrations of BOLA1 or BOLA3. CD spectra were recorded anaerobically at 21°C in 1 mm cuvettes.

## Affinity measurements using microscale thermophoresis (MST)

The bait proteins were fluorescently labeled using the Monolith NT Protein Labeling Kit RED (Nano-Temper Technologies) with NT-647 dye as recommended by the manufacturer. Labeled bait proteins (200 nM) were titrated with the indicated unlabeled proteins (from 200 μM to 6.1 nM) in buffer T (50 mM $KP_i$, pH 7.4, 150 mM NaCl, 5% glycerol, 0.05 mg/ml BSA, 0.05% Tween20). Binding assays were performed using Monolith NT.115 (NanoTemper Technologies) at 21°C (LED power between 40% and 60%, IR laser power 75%) in standard capillaries under anaerobic conditions at 680 nm. The results were processed by NanoTemper Analysis 1.2.009 and GraphPadPrism5 software to estimate $K_d$ values.

## NMR experiments

All NMR experiments required for resonance assignment and structure calculations of BOLA1 (PDB ID 5LCI; BMRB 34013) and BOLA3 (PDB ID 2NCL; BMRB 26031) were recorded on Bruker AVANCE 500, 700 and 900 MHz spectrometers on 0.5–1 mM $^{13}C,^{15}N$-labeled or $^{15}N$-labeled BOLA1 and BOLA3 samples in 50 mM phosphate buffer, pH 7.0, 5 mM DTT, containing 10% (v/v) $D_2O$. All NMR spectra were collected at 298 K, processed using the standard BRUKER software (Topspin) and analyzed through the CARA program. The $^1H$, $^{13}C$ and $^{15}N$ resonance assignment of BOLAs were performed following a standard triple-resonance and TOCSY-based approach. Secondary structure analysis was performed by TALOS+. Structure calculations of BOLA3 and BOLA1 were performed with the software package UNIO (ATNOS/CANDID/CYANA). The 20 conformers with the lowest residual target function values were subjected to restrained energy minimization in explicit water with the program AMBER 12.0. (D.A. Case et al. University of California, San Francisco). The quality of the structures was evaluated using the Protein Structure Validation Software suite (PSVS) and the iCING program. $^{15}N$ heteronuclear relaxation experiments on $^{15}N$-labeled samples of BOLA1 and BOLA3 were recorded on Bruker AVANCE 500 MHz spectrometer at 298 K to measure $^{15}N$ backbone longitudinal ($R_1$) and transverse ($R_2$) relaxation rates, as well as the heteronuclear $^{15}N[^1H]$ NOEs. The rotational correlation time values were estimated from the $R_2/R_1$ ratio using the program QUADRATIC_DIFFUSION. The relaxation data for those NHs having an exchange contribution to the $R_2$ value or exhibiting large-amplitude fast internal motions were excluded from the analysis. Theoretical estimates of the rotational correlation time under the chosen experimental conditions of magnetic field and temperature were obtained using HYDRONMR program following a standard procedure (*Figure 6—figure supplement 3*).

The interaction between apo-GLRX5 and BOLA proteins was investigated by $^1H$-$^{15}N$ HSQC NMR spectra, titrating $^{15}N$-labeled apo-GLRX5 with unlabeled BOLA1 or BOLA3, and $^{15}N$ labeled BOLA1 or BOLA3 with unlabeled apo-GLRX5 in degassed buffer N containing 10% (v/v) $D_2O$) at 298K. Spectral changes were monitored upon the addition of increasing amounts of the unlabeled partner.

Protein interaction between [2Fe-2S] holo-GLRX5 and BOLA proteins was investigated comparing $^1$H-$^{15}$N HSQC NMR spectra of $^{15}$N-labeled BOLA1 (or BOLA3) with that of a 1:1 $^{15}$N-labeled BOLA1 (or BOLA3)-unlabeled apo GLRX5 mixture chemically reconstituted with [2Fe-2S], or of $^{15}$N-labeled apo-GLRX5 with that of a 1:1 $^{15}$N-labeled apo-GLRX5-unlabeled BOLA1 (or BOLA3) mixture chemically reconstituted with [2Fe-2S], in degassed buffer N containing 10% (v/v) D$_2$O at 298K. NMR data were analyzed with CARA program.

## Miscellaneous methods

The following published methods were used: manipulation of DNA and PCR (*Sambrook and Russell, 2001*); transformation of yeast cells (*Gietz and Woods, 2002*); isolation of yeast mitochondria and post-mitochondrial supernatant (*Diekert et al., 2001*); immunostaining (*Harlow and Lane, 1998*); determination of enzyme activities and of promoter strength using *FET3*-GFP or *FIT3*-GFP reporter plasmids (*Molik et al., 2007*); in vivo labeling of yeast cells with $^{55}$FeCl (ICN) and measurement of $^{55}$Fe-incorporation into Fe/S proteins by immunoprecipitation and scintillation counting (*Molik et al., 2007*; *Pierik et al., 2009*). Antibodies were raised in rabbits against recombinant purified proteins. Antibodies against c-Myc or HA were obtained from Santa-Cruz, protein A sepharose from GE Healthcare.

## Acknowledgements

We thank Dr. DR Winge for stimulating discussions and expression plasmids containing *SDH1* and *SDH2* genes. We gratefully acknowledge the contribution of the Core Facility 'Protein Spectroscopy'. RL acknowledges generous financial support from the Deutsche Forschungsgemeinschaft (SFB 987 (also to UM), SPP 1710 and SPP 1927), LOEWE program of state Hessen, and von Behring-Röntgen Stiftung. LB has been supported by iNEXT, project number 653706, funded by the Horizon 2020 program of the European Commission, and by Instruct, part of the European Strategy Forum on Research Infrastructures (ESFRI), and supported by national member subscriptions. We thank the EU ESFRI Instruct Core Centre CERM-Italy.

## Additional information

### Funding

| Funder | Grant reference number | Author |
| --- | --- | --- |
| Deutsche Forschungsgemeinschaft | SFB 987 | Ulrich Mühlenhoff Roland Lill |
| European Commission | iNEXT 653706 | Lucia Banci |
| European Strategy Forum on Research Infrastructures | Instruct | Lucia Banci |
| Deutsche Forschungsgemeinschaft | SPP 1927 | Roland Lill |
| LOEWE program of state Hesse, Germany | Synmikro | Roland Lill |
| Deutsche Forschungsgemeinschaft | SPP 1710 | Roland Lill |
| Von-Behring-Röntgen-Stiftung | | Roland Lill |

The funders had no role in study design, data collection and interpretation, or the decision to submit the work for publication.

### Author contributions

MAU, VN, BDW, SC-B, UM, Conception and design, Acquisition of data, Analysis and interpretation of data, Drafting or revising the article; FS, MRS, LG, Acquisition of data; LB, RL, Conception and design, Analysis and interpretation of data, Drafting or revising the article

## Author ORCIDs

Roland Lill, http://orcid.org/0000-0002-8345-6518

## Additional files

### Supplementary files

• Supplementary file 1. Yeast strains and plasmids used in this study. (A) Yeast strains used in this study. Gene disruptions and promoter exchanges were generated by PCR-based gene replacement and verified by PCR as described previously (*Gueldener et al., 2002; Mühlenhoff et al., 2002*). Yeast cells were transformed by the lithium acetate method (*Gietz and Woods, 2002*). In some cells, the *TRP1* gene was disrupted by a *natNT2* cassette (*Janke et al., 2004*) in order utilize plasmids with the *TPR1* marker. (B) Plasmid constructs used in this study. The plasmids were constructed inserting the indicated genes into vector. The amino acid residues of the encoded proteins and the hexa-histidinyl tag (6xHis) are indicated.

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
