## [Decision Letter]

Thank you for submitting your article "Mitochondrial Bol1 and Bol3 function as assembly factors for specific iron-sulfur proteins" for consideration by *eLife*. Your article has been reviewed by three peer reviewers, one of whom, Klaus Pfanner, is a member of our Board of Reviewing Editors, and the evaluation has been overseen by Michael Marletta as the Senior Editor.

The reviewers have discussed the reviews with one another and the Reviewing Editor has drafted this decision to help you prepare a revised submission.

Summary:

The manuscript authored by Uzarska et al. concerns BOL-type proteins of the mitochondria. The analysis reported focuses on two approaches – genetic analysis in *S. cerevisiae* (BOL1 and BOL3) and biophysical analysis of human proteins (BOLA1 and BOLA3). This is the first analysis of the yeast genes. The authors show that a single deletion of either Bol1 or 3 produces very mild or no phenotype in the activities of target proteins. The double deletion produces a measurable phenotype which they state is milder compared with other disruptions earlier in the Fe-S pathway. They investigated the interaction of Bol1 and Bol3 with GRX5 and NFU1, two other components of the late Fe-S assembly pathway.

Overall, the experiments are of high quality and the clear linkage to Fe-S cluster biogenesis and the apparent specificity for 4Fe-4S clusters is important new information.

Essential revisions:

1) The functional difference between Bol1 and Bol3 is difficult to depict. Deletion of only one of them showed hardly phenotypical consequences, yet deleting both showed drastic effects. Is there a compensatory effect in terms of protein levels of Bol1 and Bol3 and even Nfu1 and Grx5 in bol1-δ, bol3-δ and nfu1-δ strains? Overexpression of Bola3 in bola1-δ may overcome the weak interaction of Bola3-Glrx5 to substitute for Bola1.

The observation that Bol1 and Bol3 interact with different known components of the Fe-S cluster biosynthetic pathway does not suggest to me that they have overlapping functions as claimed. Rather I would say that the double deletion produces a synthetic sick strain, consistent with both proteins acting in the same pathway. They also claim that yeast is a good model for the human disease, but in humans mutations in BOLA3 alone clearly produce a very severe biochemical and clinical phenotype, so I would be tempted to conclude that BOLA1 and BOL3 have non-overlapping functions in humans.

2) A key aspect of the yeast analysis that is missing is analysis of the levels of the proteins in cells being analyzed, particularly Bol1 and Bol3, but also Nfu1. This information is important for interpreting the differences in the phenotypes of deletion strains. But it is even more so in the cases where additional copies are expressed from plasmids. The effect of such additional copies can be interpreted if the fold overexpression is known and the relative level of protein being overexpressed to the level normally present of the protein.

At a minimum the authors need to intellectually address this issue and be more transparent about what they do and don't know. Of course, the paper would be much stronger if protein levels were measured using specific antibodies or tagged proteins that were demonstrated to be functional.

3) The relationship between the yeast and human sections of the manuscript is not so clear.

What is the sequence relationship between the yeast mBols and the human BOLAs? It appears from the alignment in Figure 5—figure supplement 2 that yeast Bol1 is more similar to BOLA1 and yeast Bol3 is more similar to BOLA3. But this is not at all clear as one is reading through the manuscript. It would be very helpful to the reader if this be addressed directly up front at the beginning of the manuscript. Also a bit of background as to what are the core features that "define" a "BOL" protein and what in general is known about "distinguishing characteristics" among members.

---

## [Author Response]

*Essential revisions:*

*1) The functional difference between Bol1 and Bol3 is difficult to depict. Deletion of only one of them showed hardly phenotypical consequences, yet deleting both showed drastic effects. Is there a compensatory effect in terms of protein levels of Bol1 and Bol3 and even Nfu1 and Grx5 in bol1-δ, bol3-δ and nfu1-δ strains? Overexpression of Bola3 in bola1-δ may overcome the weak interaction of Bola3-Glrx5 to substitute for Bola1.*

This is a valid point which we possibly have not outlined well in our manuscript. We do not see compensatory effects between Bol1, Bol2, Nfu1, and Grx5. We have already shown in the previous version of our manuscript (Figure 4) that Nfu1 overexpression cannot compensate for Bol1,3, and Bol3 does not for Nfu1. The same is true for overexpressed Grx5 that cannot compensate for the mBols (Figure 2). As a further analysis into this direction and as suggested by the Reviewers, we now have analyzed the levels of Bol1, Bol3 (see legend to Figure 9), and Nfu1 in the various yeast deletion cells to check the levels of the residual proteins. As seen in Figure 9, we do not find major compensatory changes for these ISC proteins. The level of Bol1 in *bol3Δ* cells is even lower. Hence, we find it highly unlikely that compensatory effects may explain the synthetic phenotype of *BOL1,3* deletions, or the generally rather weak effects of the deletion of the late-acting *ISC* genes. We now mention this finding in the Results in our revised manuscript. We also emphasize that the weak effects of late ISC protein deficiencies are easily explained by their rather specific requirement for maturation of only a small subset of mitochondrial Fe/S proteins (Discussion).

Author response image 1.Late ISC protein levels in various yeast strains.Wild-type (WT; BY4742), indicated *BOL* deletion strains and nfu1Δ cells were grown in (**A**) minimal medium containing 2% galactose supplemented with 50 µM ferric ammonium citrate, (**B**) iron-poor minimal medium containing 2% galactose or (**C**) lactate medium. Mitochondria were isolated and the indicated protein levels were determined by immunostaining. The Hsp70 protein Ssc1 served as a loading control. As shown below (Figure 10), wild-type levels of Bol3 were not detectable by our antibody precluding analysis of this protein.**DOI:**
http://dx.doi.org/10.7554/eLife.16673.027

*The observation that Bol1 and Bol3 interact with different known components of the Fe-S cluster biosynthetic pathway does not suggest to me that they have overlapping functions as claimed. Rather I would say that the double deletion produces a synthetic sick strain, consistent with both proteins acting in the same pathway.*

We agree with the latter part of this interpretation. Our *BOL* deletions strains clearly show that the presence of one Bol protein can suppress most or all phenotypes of the double deletion. In our view, two biochemical models may explain this situation. i) The two Bol proteins assist the same reaction. Hence, absence of only one Bol protein does not cause a major phenotype. The subtle differences between the *bol1Δ* (no phenotypes) and *bol3Δ* (occasional weak phenotypes) strains may then be due to minor target apoprotein specificities of the two Bol proteins or the different protein interactions as mentioned by the Reviewers. We emphasize, however, that we have no phenotypical evidence that the Bol protein complexes have an in vivorelevance. ii) The Bol proteins act consecutively in two independent maturation steps that both can be bypassed to some extent. Even though we find the second possibility biochemically less attractive, it is hard at this stage of analysis to more precisely define the specific molecular function of the Bol proteins. To better explain our interpretation of the relative function of Bol1-Bol3, we have improved our Discussion (first paragraph), and hope this point is much clearer now.

*They also claim that yeast is a good model for the human disease, but in humans mutations in BOLA3 alone clearly produce a very severe biochemical and clinical phenotype, so I would be tempted to conclude that BOLA1 and BOL3 have non-overlapping functions in humans.*

Apparently, our comment that yeast is a good model for human disease was found misleading. What we meant to say (and we still think is correct), is that the Bol proteins in both yeast and human cells have a major function in lipoate synthase maturation. This is the major and strongest phenotype in both yeast and humans. We have improved our statements in the Discussion to make this issue clearer. Since the two human BOLA proteins have not been analyzed in a direct comparative in vivo study, we agree with the Reviewers that any further speculation about the similarity between yeast and human Bol proteins is not justified at this stage. We have screened our manuscript for possible confusing comments and have amended them.

*2) A key aspect of the yeast analysis that is missing is analysis of the levels of the proteins in cells being analyzed, particularly Bol1 and Bol3, but also Nfu1. This information is important for interpreting the differences in the phenotypes of deletion strains. But it is even more so in the cases where additional copies are expressed from plasmids. The effect of such additional copies can be interpreted if the fold overexpression is known and the relative level of protein being overexpressed to the level normally present of the protein.*

*At a minimum the authors need to intellectually address this issue and be more transparent about what they do and don't know. Of course, the paper would be much stronger if protein levels were measured using specific antibodies or tagged proteins that were demonstrated to be functional.*

This is a valid point and connected to point 1. Protein levels in yeast are available from systematic studies (see the SGD website http://www.yeastgenome.org/). 10-fold lower amounts of Bol1 and Bol3 were found compared to Nfu1 and Grx5 (two tables were provided for the reviewers’ consideration). However, these data sets may be of limited use, because the protein values differ more than tenfold between the different published data sets. Even after normalization to one protein (e.g., Nfs1) the relative protein levels in the different data sets differ substantially. Thus, we cannot rely on published numbers. We therefore have quantitated the levels of Bol1, Bol3, and Grx5 as suggested by the reviewers (Figure 10). The analysis shows that Grx5 is (at least) 4-fold more abundant than the Bol proteins. From immunoblots we know that overexpression usually generates 10-20-fold higher levels. However, as discussed above from the analysis of protein levels in various deletions strains, the mere protein level increases do not have major effects on the phenotypes studied in our investigation. We have discussed this issue briefly. An analysis for Nfu1 was not feasible because we could not obtain purified yeast Nfu1.

Author response image 2.Protein levels of Bol1, Bol3 and Grx5 in yeast mitochondria.Wild-type (WT; BY4742) and *bol13Δ* cells were grown in minimal medium containing 2% galactose supplemented with 50 µM ferric ammonium citrate and used for isolation of mitochondria (mito). Mitochondrial proteins and the indicated amounts of purified (**A**) Bol1, (**B**) Bol3, and (**C**) Grx5 were analyzed by immunostaining with specific antibodies. Densitometry of the bands was used to calculate the proteins levels (in ng per mg mitochondria). Since our antibody did not detect wild-type amounts of Bol3 in mitochondria (**B**), only an upper limit of Bol3 could be calculated. The different gel mobility of purified and mitochondrial Bol1 and Grx5 is explained by a His-tag on purified proteins.**DOI:**
http://dx.doi.org/10.7554/eLife.16673.028

*3) The relationship between the yeast and human sections of the manuscript is not so clear.*

*What is the sequence relationship between the yeast mBols and the human BOLAs? It appears from the alignment in Figure 5—figure supplement 2 that yeast Bol1 is more similar to BOLA1 and yeast Bol3 is more similar to BOLA3. But this is not at all clear as one is reading through the manuscript. It would be very helpful to the reader if this be addressed directly up front at the beginning of the manuscript. Also a bit of background as to what are the core features that "define" a "BOL" protein and what in general is known about "distinguishing characteristics" among members.*

We realize that our general introduction into the Bol proteins was not optimal. As suggested, we have rewritten part of our Introduction (second paragraph) to better explain the various BOLA protein family members. We also added a cartoon (new Figure 1—figure supplement 1) to highlight the specific sequence features of the prokaryotic and various eukaryotic BOLA family members. We thank the reviewers for pointing out this didactic short-coming.